# Pluri-decadal (1955–2014) evolution of glacier–rock glacier transitional landforms in the central Andes of Chile (30–33ºS)

**S. Monnier[1], C. Kinnard[2]**

[1]{Instituto de Geografía, Pontificia Universidad Católica de Valparaíso, Valparaíso, Chile}

[2]{Département des Sciences de l'Environnement, Université du Québec à Trois-Rivières, Trois-Rivières, Québec, Canada}

Correspondence to: S. Monnier (sebastien.monnier.ucv@gmail.com)

## Abstract

Three glacier–rock glacier transitional landforms in the central Andes of Chile are investigated over the last decades in order to highlight and question the significance of their landscape and flow dynamics. Historical aerial photos and Geoeye satellite images were used together with common processing operations including imagery orthorectification, digital elevation model generation and cross-correlation image matching. At each site the rock glacier morphology, thermokarst area, horizontal surface displacements and elevation changes were mapped over the period 1955–2014. The evolution of the landforms over the study period is remarkable, with rapid landscape changes, horizontal surface displacements up to more than 3 m yr$^{-1}$, and elevation changes up to more than ± 1 m yr$^{-1}$. Two major landscape evolution trends are distinguished, which depend on the landform settings and associated topoclimatic conditions: (i) in areas with cold-casting conditions, the rock glacier morphology and associated cohesive mass flow have progressed at the expense of the debris-covered glacier areas, stabilizing the surface; (ii) in areas with less favourable conditions, the debris-covered glacier morphology has downwasted because of ice melt, destabilizing the surface. Two of the studied landforms initially (prior to the study period) developed from an alternation between glacial advance and rock glacier development phases. The other landform is a small debris-covered glacier having transformed into a rock glacier over the last half-century. We suggest that morphological and dynamical interactions between glaciers and

permafrost and their resulting hybrid landscapes may enhance the resilience of the mountain cryosphere against climate change.

**Key words:**

Rock glacier; Debris-covered glacier; Cryosphere landscape evolution; Flow dynamics; Remote Sensing

# 1    Introduction

Glacier−rock glacier interactions related to Holocene glacier fluctuations (e.g., Haeberli, 2005) and the current evolution of small debris-covered glaciers having survived to the post-Little Ice Age (LIA) warming (e.g., Bosson and Lambiel, 2016) are important issues in high mountain studies. They may provide key insights into the mechanisms of rock glacier development (Dusik et al., 2014) and of cryosphere stability and resilience against climate changes; the latter topic is of societal importance in arid−semiarid mountain areas, where the potential permanence of underground solid water resources subsequent to deglaciation may constitute a non-negligible water resource (e.g., Rangecroft et al., 2013).

The most striking geomorphological expression of glacier−rock glacier interactions are large glacier−rock glacier transitional landforms which are assemblages of debris-covered glaciers in their upper part and rock glaciers in their lower part (e.g., Kääb et al., 1997; Krainer and Mostler, 2000; Ribolini, 2007; Monnier et al., 2014; Janke et al., 2015). Here, it is important to recall and highlight the differences between both types of landforms (Nakawo et al., 2000; Kääb and Weber, 2004; Cogley et al., 2006; Haeberli et al., 2006; Degenhardt, 2009; Benn and Evans, 2010; Berthling, 2011). Rock glaciers are perennially-frozen homo- or heterogeneous ice−rock mixtures covered with a continuous and several metres thick ice-free debris layer that thaws every summer (known as the permafrost 'active layer'); rock glaciers movement is governed by gravity-driven permafrost creep. Debris-covered glaciers are glaciers covered with a thin (no more than several decimetres thick) and generally discontinuous debris layer; debris-covered glaciers movement is governed by gravity-driven ice creep and sometimes basal slip in response to a mass balance gradient; debris-covered glaciers do not require permafrost conditions. Rock glaciers and debris-covered glaciers exhibit distinct morphologies that are of critical importance in the surface energy balance and

subsurface heat transfer. On their surface, rock glaciers exhibit "the whole spectrum of forms created by cohesive flows" (Barsch, 1992, p. 176) of "lava-stream-like (…) viscous material" (Haeberli, 1985, p. 92). These features vary upon the case and study area; according to our field surveys in the Andes, they can be grouped in three main types: small-scale (<1 m high) ripples or undulations resulting from deformations in the active debris layer moving together with the underlying perennially-frozen core; medium-scale (1−5 m high) ridge-and-furrow assemblages resulting from the compression in the whole ice−debris mixture; and large scale (5−20 m thick and >100 m long) superimposed flow lobes upon which the first two types may naturally appear. Hereafter, we will simply refer to these features as 'cohesive flow-evocative features'. Contrarily, debris-covered glaciers are characterized by a chaotic distribution of features evocating surface instability such as hummocks, collapses, crevasses, meandering furrows, and thermokarst depressions and pounds. As a consequence, on rock glaciers the large- and fine-scale surface topography is rather smooth and convex, whereas on debris-covered glaciers it is rather rough and concave. Another morphological difference is the presence of ice visible from the surface: whereas ice is generally invisible from the surface of rock glaciers, it is frequently exposed on debris-covered glaciers due to the discontinuity of the debris cover or the occurrence of the aforementioned morphological features. Finally, and correlatively, over pluri-annual to pluri-decadal periods the morphology of well-developed rock glaciers is stable (beside cases of climate warming-related destabilizations, the geometry of surface features evolves but their overall pattern remains the same) while debris-covered glacier morphology is characterized by instability (surface features rapidly appear and disappear).

That being said, according to the literature at least three types of glacier−rock glacier interactions can be distinguished:

(i) The readvance(s) and superimposition/embedding of glaciers or debris-covered glaciers onto/into rock glaciers, with related geomorphological and thermal consequences (Lugon et al., 2004; Haeberli, 2005; Kääb and Kneisel, 2006; Ribolini et al., 2007 and 2010; Bodin et al., 2010; Monnier et al., 2011; Monnier et al., 2014; Dusik et al., 2015). This is the *stricto sensu* significance of 'glacier−rock glacier relationships' (Haeberli, 2005) as defined by what has been called the 'permafrost school' in reference to the long-term 'rock glacier controversy' (see Berthling, 2011).

(ii)   The continuous derivation of a rock glacier from a debris-covered glacier by evolution of the surface morphology (see above) together with the conservation and creep of a massive and continuous core of glacier ice (e.g., Potter, 1972; Johnson, 1980; Whalley and Martin, 1992; Potter et al., 1998; Humlum, 2000). This process was not initially called a 'glacier–rock glacier relationship'; this view is indeed held by what has been called the 'continuum school' in opposition to the permafrost school (Berthling, 2011). Nevertheless, such phenomenon does belong, literally, to the domain of glacier–rock glacier interactions.

(iii)  The transformation of a debris-covered glacier into a rock glacier not only by the evolution of the surface morphology but also by the evolution of the inner structure, i.e., transformation of the debris-covered continuous ice body into a perennially frozen ice–rock mixture by addition from the surface of debris and periglacial ice and fragmenting of the initial glacier ice core. This has been described as an alternative to the dichotomous debate between the permafrost school and continuum school (Monnier and Kinnard, 2015); such phenomenon has been described as achievable over human life or historical time scale (Schroder et al., 2000; Monnier and Kinnard, 2015; Seppi et al., 2015).

In the present study, we aim to provide insights into the aforementioned issue using the variety of glacier–rock glacier transitional landforms encountered in the semiarid Andes of Chile and Argentina. These landforms have shown a particularly rapid evolution over the last decades which allow studying glacier–rock glacier interactions on an historical time scale. Three landforms with distinct morphologies have been chosen in the central Andes of Chile in an attempt to diagnose their geomorphological significance, especially in terms of glacier–rock glacier interactions and cryosphere persistence in the current climatic context. To this purpose, this study makes use of aerial and satellite imagery as well as remote sensing techniques in order to document the morphological and dynamical evolution of the studied landforms over a pluri-decadal time span.

## 2   Study sites

We studied three glacier–rock glacier transitional landforms in the central Andes of Chile, respectively named Navarro, Presenteseracae, and Las Tetas (Fig. 1). Navarro and Presenteseracae are located in the Navarro valley, in the upper Aconcagua River catchment

(33º S). Las Tetas is located in the Colorado valley, in the upper Elqui River catchment (30º S).

## 2.1 Upper Navarro valley

The upper Navarro valley belongs to the Juncal River catchment and Juncal Natural Park, which are part of the upper Aconcagua River catchment, in the Valparaíso Region of Chile (32°53' S, 70°02' W; Fig. 1). In the Juncal catchment (~1400−6110 m asl), glaciers cover 14% of the area (Bown et al., 2008; Ragettli et al., 2013) while active rock glaciers cover almost 8% (Monnier and Kinnard, 2015). The climate is a mediterranean mountain climate. Brenning (2005) and Azócar and Brenning (2010) located the 0ºC isotherm of mean annual air temperature (MAAT) close to 3700 m asl and defined precipitations above 3000 m asl as ranging between 700 and 800 mm yr$^{-1}$. An automatic weather station located at 2800 m asl at the foot of the Juncal glacier, 10 km SW from Navarro valley, recorded a MAAT of 6.3ºC during the hydrological year 2013−2014. The upper Navarro valley crosses, from west to east, the Albánico formation (Upper Cretaceous; andesites, volcanic breccias), the San José formation (Lower Cretaceous; limestones), and the Lagunilla formation (Upper Jurassic; sandstones, lutites, gypsum). The glacial footprint is conspicuous through the Navarro valley: the valley is U-shaped, with corries in the upper parts and latero-frontal moraines in the lower parts (Fig. 2 and 3).

### 2.1.1 Navarro

Navarro fills the major part of the upper Navarro valley floor between ~3950 and 3450 m asl (Fig. 3). The landform was described by Janke et al. (2015, p. 117) as a system composed of several classes of debris-covered glaciers and rock glaciers according to their presumed ice content. It is indeed a huge (>2 km long and up to >1 km wide) and complex assemblage with debris-covered glacier morphology in its upper parts and rock glacier morphology in its lower parts. The main presumed flow direction of the landform points towards N170º. At least ten conspicuous and sometimes >15 m high morainic crests are visible at the surface of the landform, some of them being included in the rock glacier morphological unit. At one location (red circle in Fig. 3), the superposition of two series of morainic crests onto a rock glacier lobe suggests that the landform developed from a succession of glacier advances and rock glacier development phases.

Navarro is divided between an eastern and a western unit; the two being separated by a central series of aligned morainic crests (Fig. 3). The eastern unit, which is located in the more shadowed north-eastern part of Navarro valley, is ~1.2 km long, and about two thirds of its area exhibits a rock glacier morphology. The terminal part exhibits three adjacent terminal lobes. The western unit is ~2.4 km long and more complex. Sets of embedded morainic crests in the upper part delimit the retreat of a former glacier. The median part (~1 km long) is peculiar, with the boundary between the debris-covered and rock glacier morphology extending far downslope and following the contour of an elongated central depression (10–15 m lower in altitude than the lateral margins) (Fig. 3 and 4). This central depression is characterized by numerous and large (up to 50 m of diameter) thermokarst depressions with bare ice exposures, generally on their south-facing walls. The lower part of the western unit exhibits a rock glacier morphology and three superimposed fronts close to the terminus, the slope of the lowest front being gentler than that of the two upper fronts, which are almost at the same location.

Monnier and Kinnard (2015) provided an empirical model of permafrost probability based on logistical regression for the upper Aconcagua River catchment. According to this model, Navarro may be in a permafrost state. The permafrost probability is close to 1 in the upper parts, nevertheless there is a marked decreasing gradient in permafrost probability from 0.9 to 0.7 between the central part and the terminus of the western unit.

## 2.1.2 Presenteseracae

Presenteseracae is a small (~600 m long and 300 m wide) debris-covered glacier located between ~4080 and 3800 m asl, in a narrow, SW-facing cirque, ~300 m above and only 500 m east of Navarro (Fig. 3). The main presumed flow direction points towards N225°. This landform has been thoroughly analysed by Monnier and Kinnard (2015). The debris-covered glacier exhibits rock glacier features in its lower part (see also Fig. 4). The transverse and curved ridges (<1.5 m high) and well-defined steep frontal talus slopes (~10 m high) have appeared during the last 15 years. The permafrost model of Monnier and Kinnard (2015) gave a permafrost probability of 1 for the whole Presenteseracae landform. The authors also correlated the development of the cohesive flow-evocative rock glacier morphology with the low estimated sub-debris ice ablation rates, and demonstrated that the sediment store on Presenteseracae and the potential formation times are in agreement with common rock wall retreat rates. They concluded that Presenteseracae is a debris-covered glacier currently

evolving into rock glacier. In the upper part of the landform, the debris cover is very thin (a few cm) and bare ice exposures are frequent. The debris cover thickens to more than 60 cm in the lower part, where the rock glacier morphology develops below a steeper sloping segment. Push moraine ridges (Benn and Evans, 2010) occur at the surface above 3780 m asl (Fig. 3). The lower part which displays a rock glacier morphology is clearly composed of two adjacent lobes, dividing away from a morainic crest overridden by the landform (Fig. 4). Depressed meandering furrows where buried ice is exposed are also present. During hot summer days the water flowing in the northernmost furrow sinks down a hole just before the front.

## 2.2   Las Tetas

Las Tetas is located in the Colorado valley, which is the uppermost part of the Elqui River valley, in the Norte Chico Region of Chile (30°10' S, 69°55' W; Fig. 1). Elevations in the Colorado valley range between ~3100 m asl and 6255 m asl. The landform is located on the south-facing side of Cerro Las Tetas (5296 m asl), less than one km south of Glacier Tapado (e.g., Ginot et al., 2006; Pourrier et al., 2014). The climate of the area is a semiarid mountain climate. At the La Laguna artificial dam (~3100 m asl, 10 km west of the study site), the mean annual precipitation was 167 mm during the 1970–2009 period, and the mean annual air temperature was 8°C during the 1974–2011 period. The 0°C-isotherm is located near 4000 m asl (Brenning, 2005; Ginot et al., 2006). Materials composing the rock basement belong to the Pastos Blancos formation (Upper Palaeozoic; andesitic to rhyolitic volcanic rocks). A set of embedded latero-frontal moraines is encountered ~700 m downslope from the front of Las Tetas, between ~4170 and 4060 m asl.

Las Tetas is a ~1 km long landform located between 4675 and 4365 m asl (Fig. 5). The main presumed flow direction points towards N140º. The boundary between debris-covered and rock glacier morphology is clear, in the form of a large and deep furrow, and divides the landform in two approximately equal units. The upper unit is characterized by a chaotic and hummocky morphology, vast (up to more than 50 m of diameter) and deep (up to 20 m) thermokarst depressions exposing bare ice generally along their south-facing walls. The lower part of the landform exhibits tension cracks superimposed onto the ridge-and-furrow pattern. The front of Las Tetas is prominent; its talus slope, which may bury sediments or outcrops downward, is almost 100 m high (Fig. 4). According to the logistic regression-based empirical permafrost model proposed by Azócar (2013) for the area, the 0.75 probability level crosses the landform in its central part (Fig. 5). Permafrost favourability index (PFI) values proposed

by Azócar et al. (2016a and b) are >0.7 in the upper part and between 0.6 and 0.7 in the lower part.

## 3 Material and methods

### 3.1 Satellite image and aerial photo processing

We searched for and acquired historical aerial photos and more recent satellite images for the three study sites. Stereo pairs of aerial photos were inspected, selected, and scanned at the Geographic and Military Institute (IGM) of Chile. Scanning was configured in order to yield a ground resolution of 1 m. At Las Tetas, photos from 1956 and 1978 were selected; at Navarro and Presenteseracae (Navarro valley), photos from 1955 and 2000 were selected. A stereo pair of Geoeye satellite images was also acquired for each site. The Geoeye imagery was acquired on 23 March 2012 and 14 February 2014 at Las Tetas and Navarro valley, respectively, as panchromatic image stereo pairs (0.5 m of resolution) along with four bands in the near-infrared, red, green, and blue spectra (2 m of resolution).

Orthoimages, orthophotos, and altimetric information were generated from the data. The first step involved building a digital elevation model (DEM) from the stereo pair of Geoeye satellite images. The Geoeye images were triangulated using a Rational Polynomial Camera (RPC) model supplied by the data provider. The exterior orientation was constrained using one or two (according to the site) ground control points (GCPs) acquired with a differential GPS system in the field in 2014 over bedrock outcrops visible on the images. Sets of three-dimensional (3D) points were extracted automatically using standard procedures of digital photogrammetry (Kääb, 2005) and edited manually to remove blunders. A $2 \times 2$ m DEM was generated using triangular irregular network (TIN) interpolation of the 3D points. The same processing scheme was followed for the aerial photo stereo pairs using control points visible both on the Geoeye image and the aerial photo stereo pairs. The vertical bias of the aerial photo DEMs was calculated by comparison with the Geoeye DEMs over flat and stable areas outside the landform studied and was removed from the subsequent calculations (see below). The automatic and manual extraction of 3D points from aerial photo stereo pairs proved to be challenging in steep areas with unfavourable viewing geometry. The process failed for the 1955 stereo pair of Navarro valley, with only a very sparse set of 3D points extracted and including possible blunders, ruling out the possibility to generate a reliable and complete DEM and to estimate the vertical bias.

The Geoeye images were pansharpened and orthorectified using the Geoeye DEM. The aerial

photos were then orthorectified using the corresponding DEMs, except when no reliable DEM

could be obtained (as for 1955 at Navarro); in that case the Geoeye DEM was used. The

orthorectification was constrained by the internal camera information, tie points, and ground

control points (GCPs) extracted during the process. The accuracy of the orthorectification was

estimated using the GCPs. The root mean square error (RMSE) corresponding to the sets of

GCPs at the different times is displayed in Table 1. The ground resolution of the orthophotos

was then resampled at 0.5 m in order to equal that of the Geoeye products.

The altimetric information was used to calculate the elevation changes of the landforms

between the different dates (after removal of the vertical bias). The total elevation change was

further converted in annual rates of elevation change. As outlined by Lambiel and Delaloye

(2004), elevation changes at the surface of rock glaciers may be explained by several and

possibly concomitant factors: downslope movement of the landform and advection of local

topographic features, extensive or compressive flow, and melting or aggradation of internal

ice. Therefore, it is difficult to unambiguously interpret elevation changes. Studying the

Muragl rock glacier (Swiss Alps), Kääb and Vollmer (2000) highlighted how mass advection

caused subtle elevation changes (between $-0.20$ and $+0.20$ m yr$^{-1}$), while surface lowering of

up to $-0.5$ m yr$^{-1}$ were considered as indicative of massive losses of ice. Accordingly, taking

into account the range of values measured and the uncertainty (or detection threshold) on the

measurements (see Table 2), we used an absolute value of 0.50 m yr$^{-1}$ to generally

discriminate between 'moderate' and 'large' vertical changes. The former were considered to

relate primarily to the downslope expansion of the landform (including long profile adaptation

and advection of topographic features) and, thus, to extensive flow; in the case of the latter,

additional ice melting or material bulging by compression were considered necessary in the

interpretation.

**3.2  Image interpretation**

The geomorphology of each landform was carefully interpreted from the orthoimages and

orthophotos. First, we located and mapped the boundary between debris-covered and rock

glacier morphology, according to the detailed criteria of differentiation presented in the

Introduction section of this work. The thermokarst area was also monitored over time by

mapping the thermokarst depressions at the surface of the landforms as polygonal shapes, and

1 their total area was calculated. Salient and recently appeared features such as cohesive flow-
2 evocative ridges on Presenteseracae and cracks on Las Tetas were also mapped.

## 3.3 Image feature tracking

We used a cross-correlation image matching technique in order to measure horizontal
displacements at the surface of the landforms. Cross-correlation image matching is a sub-pixel
precision photogrammetric technique that has been widely used for studying the kinematics of
glaciers, rock glaciers, and other mass movements. We followed the principles and guidelines
provided by Kääb and Vollmer (2000), Kääb (2005), Wangensteen et al. (2006), Debella-Gilo
and Kääb (2011), and Heid and Kääb (2012). The image correlation software CIAS (Kääb and
Vollmer, 2000; Heid and Kääb, 2012) was used for this purpose. CIAS uses two orthoimages
(from spaceborne, airborne, or terrestrial sensors) of the same area and resolution but at
different times. CIAS computes the normalized cross-correlation (NCC) as an estimate of the
similarity of image intensity values between matching entities in the orthoimage at time 1 ($I_1$)
and their corresponding entities in the orthoimage at time 2 ($I_2$). In $I_1$, a 'search template' is
defined around each pixel located manually or automatically inside a regular grid; the
software extracts this search template from $I_1$ and search for it in $I_2$ within the area of a
predefined search window (see Fig. 2 in Debella-Gilo and Kääb, 2011, p. 132); the algorithm
then computes the NCC coefficient between the search template in $I_1$ and the one in $I_2$ and
moves the search template until the entire search window is covered. The location that yields
the highest correlation coefficient within the search window is considered as the likely best
match for the original location in $I_1$. The size of the search template and search window is
defined by the operator. Once measurements are achieved, results are filtered (Wangensteen
et al., 2006).

For all sites we used a search template of 15 pixels, which fits the textural characteristics of
the surfaces. The search window size was defined depending upon the maximum expected
displacements (Kääb and Vollmer, 2000). The base of the front of the landform was digitized
when it was clearly identifiable on the orthoimages. Then for each time interval the maximum
front displacement was measured and used for defining the search window size accordingly:
we converted the maximum front displacement in pixels, multiplied the value by 2, and chose
an upper rounded value; hence, being sufficiently large compared to the maximum front
displacement the search window could track potential larger displacements on the surface
(Table 3). The precision of the measurement of the maximum front displacement was

estimated to be ±5 m taking into account the errors related both to orthorectification and the mapping error on the images. Before measuring displacements on the landforms the images were co-registered to one another using a polynomial transformation and by matching reference stable boulders (>1 m of diameter) or prominent rock outcrop corners between $I_1$ and $I_2$; the operation had to yield a RMSE less than or around 1 m in both $x$ and $y$ directions before being validated. The NCC algorithm was performed over the whole area of the landforms using a 5 m-spacing grid, except in the case of the 2000–2014 period at Navarro where a 10 m-spacing grid was used in order to avoid a too large amount of data. The subsequent filtering procedure excluded points that did not meet the following conditions: (i) the displacement magnitude must be higher than the RMSE generated by the orthorectification and co-registration steps (Table 2); (ii) the azimuthal deviation from the general landform flow direction must be less than 40º; (iii) the maximum cross-correlation coefficient in the search template around the point ($r_{max}$) must be higher than $\bar{r}_{max}$, the latter $\bar{r}_{max}$ being the maximum cross-correlation coefficient's average value for all measured points; (iv) the average cross-correlation coefficient in the search template must be less than $\bar{r}_{avg} + 1\sigma$, the latter $\bar{r}_{avg}$ and $\sigma$ being the average cross-correlation coefficient's average and standard deviation values, respectively, for all measured points. The use of the latter criterion aimed at removing noise data from the results, as suggested for example in the online CIAS recommendation. Eventually, remaining mismatches were manually removed after the filtering. The total displacements measured by the program were converted in annual displacement rates, and the displacement vectors were mapped.

Finally, the main streamlines at the surface of the landforms studied were interpreted and mapped from the displacement vector fields obtained. We differentiated between well-defined and moderately defined streamlines according to their continuity and regularity in heading. The streamlines were useful in helping interpreting elevation changes, i.e., deducing related mechanisms.

## 4 Results

Fig. 6, 7, and 8 show the orthophotos and orthoimages obtained at each site, together with the boundary between debris-covered and rock glacier morphology and the front slope base at each time. As mentioned in the Method section, no reliable and complete DEM could be obtained for the Navarro valley in 1955; therefore, the 1955–2000 period lacks of elevation change measurements at Navarro and Presenteseracae. On another hand, measurements of

horizontal displacements (see in Figures 9a–13b) yielded overall satisfactory results, despite subtle performance variations between sites and periods. Only for Navarro between 1955 and 2000 the results were too noisy and contained too many mismatches; they were thus discarded.

## 4.1 Navarro

At Navarro, rock glacier morphology areas have progressed spatially between 1955 and 2014, especially in the eastern unit where they now approximately represent three quarters of the total area; the boundary between debris-covered and rock glacier morphology areas has progressed upward considerably (~400 m) between 1955 and 2014 (Fig. 6). In the western unit, the spatial progression of the rock glacier morphology has been much more limited and occurred inward from the margins; the lower position of the boundary between debris-covered and rock glacier morphology has followed the overall displacement of the feature and has not progressed upward. This overall morphological evolution can be related to the reduction in thermokarst areas: between 1955 and 2000, thermokarst area expanded from 11,950 to 16,520 $m^2$, before shrinking by a factor of two in less than 15 years (8,560 $m^2$ in 2014).

Between 2000 and 2014 the landform surface advanced at horizontal velocities ranging between 0.15 and 0.99 m $yr^{-1}$ and averaging 0.49 m $yr^{-1}$ (Fig. 9a). Beside the upper and steeper north-eastern part, horizontal displacement speeds increase significantly from the upper to the lower parts of the landform; they reach their maximum values at ~3500 m asl, at the location of the two upper fronts of the western unit terminus; then, they decrease abruptly in the lower terminal lobe (Fig. 9a). The flow pattern distribution correlates with the morphology distribution. Displacement vector patterns (Fig. 9a) and corresponding stream lines (Fig. 9b) are better defined in rock glacier morphology areas than in debris-covered glacier morphology areas. Groups of landform centreline-parallel streamlines evocative of a cohesive mass flow are precisely located in the rock glacier morphology areas, both in the eastern and western unit. These streamlines correlate with moderate elevation changes (<0.5 m $yr^{-1}$ of absolute amplitude) and thus illustrate extensive flow. In the terminus part, the streamlines sometimes diverge towards the lateral margins of the landform and are associated to significant surface heaving (close to +0.5 m $yr^{-1}$); in that case they rather express compressive processes. A very conspicuous phenomenon appears in the eastern part: the divergence in flow pattern direction between the main and central area (streamlines head

towards south) and the two easternmost terminal flow lobes (streamlines head towards SSE), as if the latter were pushed from their western side by the former.

By contrast, in the debris-covered glacier areas, displacement vectors are fewer and streamlines less well defined. The movement of the debris-covered glacier is nevertheless attested by landform centreline-parallel streamlines corresponding with vast area of surface heaving between 3600 and 3750 m asl. The sparse results in the debris-covered glacier areas could be explained by the lower efficiency of the technique in these areas due to topography (rough, chaotic, and concave), variations in textural characteristics, or less spatially coherent dynamics.

Elevation changes of the surface between 2000 and 2014 are more pronounced and heterogeneous in debris-covered glacier morphology areas than in rock glacier morphology areas, and in the western unit than in the eastern unit (Fig. 9b). This highlights the transition from unstable topography to more stable topography between the former and the latter.

Nevertheless, maybe the most striking in Fig. 9a and Fig. 9b is, in the central depression of the western unit, the spatial correspondence between (i) the downward bending of the boundary between the debris-covered and rock glacier areas, (ii) the inward progression of the latter morphology, (iii) streamlines converging from the high-standing margins in rock glacier area towards the depressed landform centreline in debris-covered glacier area, and (iv) longitudinal alternation of very large surface lowering (until more than $-1$ m yr$^{-1}$) and moderate surface heaving. The most pronounced surface lowering occurs at the thermokarst locations. The concomitance of the four abovementioned phenomena clearly highlights the concomitance of general downslope movement and ice losses-related downwasting in the area where the debris-covered glacier spatially embeds into the rock glacier.

## 4.2  Presenteseracae

Monnier and Kinnard (2015) described and detailed how Presenteseracae has developed rock glacier attributes during the last decades and especially since 2000. The rock glacier morphology (continuous debris cover, frontal slope, cohesive flow-evocative ridges on the surface) has indeed appeared during the last 15 years (Fig. 7). Nowadays, the whole lower half of the landform exhibits rock glacier morphology. In the southern part of the landform, however, this morphology is less well defined and more unstable; it is conspicuously cut by a central furrow and exhibits few areas of bare ice over which debris slumps may occur. In the

northern part of the landform, the rock glacier morphology is more developed; there is neither remaining bare ice area nor evidences of debris cover instability and sliding.

The absence of thermokarst at the surface of the landform for both periods studied may be explained by the small landform size but also by the cold conditions casted by the cirque topography (permafrost probability defined by the model in Fig. 3 is 1).

Between 1955 and 2000, horizontal displacements at the surface of the landform ranged between 0.18 and 3.46 m yr$^{-1}$ and averaged 1.91 m yr$^{-1}$ (Fig. 10a). Between 2000 and 2014 period, horizontal displacements ranged between 0.15 and 2.23 m yr$^{-1}$ and averaged 1.27 m yr$^{-1}$ (Fig. 10b). Hence there is a slight decrease in velocity between 1955–2000 and 2000–2014. Displacements are one order of magnitude higher than those on the surface of Navarro and appear as relatively high for rock glaciers (see reference values in Barsch, 1996; Haeberli et al., 2006; Scapozza et al., 2014). Their amplitude correlates with the very fast landscape evolution at the site (Monnier and Kinnard, 2015). Whereas displacement speeds distribute quite heterogeneously at the surface, they have tended to homogenize since 2000. The displacement speed heterogeneity may express the fact that local sliding of the debris cover over the ice interferes with the whole creeping of the landform, due to the current debris-covered glacier–rock glacier transition. Groups of landform centreline-parallel streamlines evocative of a cohesive mass flow dispose accordingly to the topographic organisation (Fig. 11a and 11b) and correlatively to the landform downslope expansion (Fig. 7); nevertheless, they are less conspicuous and coherent than at Navarro. Between 2000 and 2014, in the lower part of the landform, they dispose perpendicularly to the cohesive flow-evocative ridges that can thus be related to the current movement. Some of these streamlines run from the upper to the lower part of the surface. In detail, one will also notice the occurrence of streamlines converging towards one another or towards depressed topographic features: either at the location of surface heaving (for both periods, see around 3900 m asl, in the upper part of the landform), which may highlight the (temporary?) formation and distal bulging by compression of individual flow lobes; or at the location of pronounced surface lowering (lower southern part between 2000 and 2014), which highlights the interference of ice losses-related downwasting with extensive flow.

Generally, the elevation changes are less contrasted in the northern than in the southern part of the landform (Fig. 11b). This correlates the more chaotic and unstable morphology identified in the latter area (see the Study site section).

## 4.3  Las Tetas

Since 1956, the boundary between debris-covered and rock glacier morphology, located in a large furrow dividing the landform in two equal parts, has only followed the overall displacement of the landform (Fig. 8, 12a, and 12b). There has not been any upward or lateral progression of the rock glacier morphology. A noticeable morphological evolution is the apparition of tension cracks in the lower part of the landform during the last decades (Fig. 5 and 8). These cracks are discernible in some locations on the 1978 orthophoto and appear fully widespread on the 2012 Geoeye image. On another hand, thermokarst, mainly but not only located in the upper part, is striking by its aspect (depressions occur in the centre of coalescent mounds, reminiscing of impact craters) and its rapid evolution (Fig. 8). Between 1956 and 1978 thermokarst depressions evolved in detail and displaced without major change in area (from 21,700 to 23,200 $m^2$); after 1978, some disappeared and were replaced by a smoother surface; their area decreased by a factor of two (11,100 $m^2$ in 2012).

The performance of cross-correlation image matching for measuring horizontal displacements was better over the more convex topography of the rock glacier morphology area, on the top of positive surface features (mounds in the debris-covered glacier area, large ridges in the rock glacier area), and, surprisingly, between 1956 and 1978 than between 1978 and 2012 (Fig. 12a and 12 b). Between 1956 and 1978, horizontal displacements at the surface of the Las Tetas landform ranged between 0.17 and 1.46 m $yr^{-1}$ and averaged 0.82 m $yr^{-1}$ (Fig. 12a). Between 1978 and 2012, they ranged between 0.12 and 1.41 m $yr^{-1}$ and averaged 0.76 m $yr^{-1}$ (Fig. 12b). There is thus a constancy in displacement speeds from one period to another. Displacement speeds are heterogeneous in the debris-covered glacier area due to the chaotic and unstable topography; they nevertheless also exhibit some spatial variability in the rock glacier area according to the topography (i.e., variations from one large-scale ridge to another). Corresponding streamlines (Fig. 13a and 13b) are remarkable for their distribution and evolution. Between 1956 and 1978, groups of landform centreline-parallel streamlines predominated and particularly linked the upper debris-covered with the lower rock glacier part (Fig. 13a). They subtly tended to diverge from the landform central flow axis at the location of large surface heaving in the terminus part, hence traducing bulging by compression, and tended to converge with one another in the upper part at the locations of thermokarst-related large surface lowering, hence traducing the concomitance of extensive flow and ice losses-related downwasting. It is nevertheless striking how after 1978 flow patterns differ much more between the upper and the lower part (Fig. 13b). Indeed, in the

upper debris-covered glacier part, beside straight flow lines on the lateral margins, most streamlines converge towards thermokarst depressions and furrows and correlate with relatively pronounced surface lowering (close to $-0.5$ m yr$^{-1}$). Conversely, in the lower rock glacier part, landform centreline-parallel streamlines still predominate, though less conspicuously and coherently than between 1956–1978. Above all, they are not anchored anymore in the upper part: the boundary between the upper debris-covered and the lower rock glacier part evolved from being purely morphological to being both morphological and dynamical. The apparition of a dense network of tension cracks in the lower part may then be explained by a dynamical de-coupling between both parts and the subsequent instability in the lower part; this is strongly supported by the subtle convergence of some streamlines towards zones of cracks or a newly formed thermokarst depression.

As for Navarro, elevation changes are more contrasted in the debris-covered glacier morphology area than in the rock glacier morphology area (Fig. 13a and 13b). They also noticeably reduced in amplitude from 1956–1978 to 1978–2012. This reduction can be seen as correlative of the reduction in thermokarst areas and corresponding topography smoothing in the upper debris-covered glacier part.

## 5   Discussion

The three cases studied have distinct significance in terms of glacier–rock glacier relationships and cryosphere persistence under ongoing climate change. Our results lead us to consider the following issues: (i) initial development of the landforms; (ii) differences between debris-covered and rock glacier areas; (iii) current evolution, especially dynamical interactions between debris-covered and rock glacier and future evolution of the landforms.

### 5.1   Initial landform development

Navarro and Las Tetas are composite landforms with a debris-covered glacier in their upper part and a rock glacier in their lower part. Considering the clear spatial organisations of surface features and the strong morphological boundaries, in particular the way the debris-covered glacier embeds into the rock glacier in the Navarro's western unit and the abrupt transition at Las Tetas, these landforms most probably result from the (re)advance(s) of glaciers onto, or in the back of pre-existing rock glaciers. Many other examples of such development of glacier–rock glacier assemblages were studied and reported in the literature (Lugon et al., 2004; Haeberli, 2005; Kääb and Kneisel, 2006; Ribolini et al., 2007 and 2010;

Bodin et al., 2010; Monnier et al., 2011; Monnier et al., 2014; Dusik et al., 2015). In the central part of the Navarro's western unit, the elevated lateral margins exhibit cohesive flow-evocative ridges, which probably resulted from the lateral compression exerted by the glacier during its advance ('composite ridges' of the glaciological terminology, Benn and Evans, 2010; p. 492). Also, the boundary between the debris-covered and rock glacier morphologies in 1955 (Fig. 6) gives a minimum indication of the lowest advance of the debris-covered glaciers onto the rock glaciers. However, the origin and age of the rock glaciers located in the lower part of the landforms are almost impossible to assess; nonetheless, considering the context, they may have developed following several glacier advances and moraine deposition phases, suggesting the idea of a cycle in the landform development (see Study site and the red circle in Fig. 3). Such development has led the rock glacier being cut off from the main rock debris sources (i.e., the rock walls up-valley), resulting in the rock glacier being dependent on the ability of the debris-covered glacier to provide material (debris and ice) required for the sustainment of the rock glacier.

Presenteseracae is a completely distinct case. As studied by Monnier and Kinnard (2015) and the present work, in 1955 Presenteseracae was a debris-covered glacier and is now a debris-covered glacier transforming into a rock glacier. The initial development phase, or in this case the 'glacier–rock glacier transformation', has been occurring over the last decades. In less than 20 years, the surface debris cover spread over almost all of the northern part; a front appeared at the terminus, and cohesive-flow evocative ridges appeared in the lower part, perpendicularly to flow streamlines. The latter ridges may be related to emergent, debris-rich shear planes (Monnier and Kinnard, 2015) bended by the landform movement. Displacement speeds were particularly high between 1955 and 2000, in agreement with the fast landscape evolution, before slowing down after 2000, which may reflect an acceleration of the transition towards a rock glacier. In the current state of our knowledge, what may have occurred in the internal structure in response to these drastic surface changes is uncertain: the continuous glacier core may however evolve into patches of buried ice progressively mixed with ice-mixed debris accumulated onto the surface, as suggested by the identification of potential nascent individual flow lobes (See subsection 4.2.).

## 5.2  Differences between debris-covered and rock glacier areas.

Our study basically relied on the landscape differentiation between debris-covered and rock glacier areas. The criteria enounced and discussed in the Introduction section have been used

to distinguish and partition the surface morphology of the landforms studied. Our subsequent results show that, at Navarro and Las Tetas, debris-covered and rock glacier areas are characterized by contrasting patterns of horizontal displacements and elevation changes. Flow patterns in rock glacier areas are conspicuous and express the cohesive downslope expansion of the landform in the direction of the main longitudinal axis; in terminal and marginal areas, significant to pronounced bulging occurs by compression. Flow patterns in debris-covered glacier areas are generally less conspicuous and express the interference of instability processes, in particular ice melt-governed downwasting, with extensive flow regime. In particular, elevation changes in debris-covered glacier areas have larger amplitudes and are spatially heterogeneous. As mentioned before, the lower density and definition of displacement vector patterns and corresponding streamlines in the debris-covered glacier areas may be explained by the inherently less cohesive mass flow. These different flow dynamics appear perfectly coherent with the definition of and distinction made between debris-covered and rock glaciers in the Introduction section.

## 5.3   Current evolution and its significance

### 5.3.1  Landscape evolution

All the landforms studied are characterized by a rapid landscape evolution over the last few decades. Changes occurred over the entire surface (Presenteseracae), in the contact/transition area between debris-covered and rock glacier and in the debris-covered glacier area (Navarro), or even in both areas though more subtly in the rock glacier area (Las Tetas). This continuum in surface evolution perhaps best illustrates the process of glacial–periglacial transition. To our knowledge, an important result of our study not previously reported is the observed upward progression of the rock glacier areas at the expense of the debris-covered glaciers on such composite landforms. At Presenteseracae, over a time span of a few decades, the rock glacier morphology has grown from being inexistent to covering approximately half the landform area. At Navarro, rock glacier areas have subtly (in the western unit) or considerably (in the eastern unit) expanded, until, in the latter case, covering most parts of the essentially debris-covered glacier morphology present initially. As a first order consideration, topoclimatic conditions seem to play a key role in this differentiated evolution: Presenteseracae and the eastern unit of Navarro are located in more shadowed and thus colder sites (see Fig. 3 and 5).

## 5.3.2 Dynamical evolution

The dynamical evolution correlates with the landscape evolution, with varying degree according to the site. This is quite logical: as stated in introduction of this work, when areas with debris-covered glacier morphology evolve into areas with rock glacier morphology, changes occur in the surface energy balance and subsurface heat transfers, which is likely to result in changes in flow dynamics depending upon the topography and the topoclimatic context. Here, we have to consider both the flow relationships between debris-covered and rock glacier areas and the flow pattern evolution related to the morphological evolution. At Navarro, debris-covered and rock glacier areas appear to be dynamically linked since streamlines connect both areas, which is particularly noticeable in the central depression of the western unit (Fig. 9b). However, upslope, the debris-covered glacier appears more dynamically disconnected from the rock glacier. In the landform's eastern unit, unfortunately the lack of displacement results for the 1955–2000 period impedes observing changes in flow patterns that may be related with the considerable upward progression of the rock glacier morphology. However, how displacement speeds regularly increase from the upper to the terminus of the eastern unit (Fig. 9a) may suggest that the spatial patterns of cohesive mass flow have progressively developed as the rock glacier upper limit was 'eating' away the debris-covered glacier features (Fig. 6).

At Presenteseracae, the flow pattern is undifferentiated over the whole landform. Nevertheless, streamlines tend to be interrupted between the lower and the upper part and the identification of potential distinct flow lobes in the upper part suggests that the debris-covered glacier ice core may be currently dynamically fragmenting (Fig. 11a and 11b). This supports the idea that transition from debris-covered towards rock glacier may proceed internally by fragmentation of the ice core and its progressive mixing with debris. On another hand, over the studied period, the overall slowing down of the landform and the trend toward speed homogenization throughout the surface (Fig. 10a and 10b) correlate well with the glacier–rock glacier transition. As suggested before, the remaining heterogeneity in surface displacements may reflect local sliding of the debris cover over the ice which interferes with the bulk displacement of the landform, implying that the glacier–rock glacier transformation is not yet achieved. Las Tetas is another peculiar case. The morphological and flow pattern evolutions highlight how the debris-covered glacier and rock glacier have been decoupling from one another since 1978, when they were still united by common streamlines. This has resulted in instability processes in the lower rock glacier part (Fig. 13a and 13b, as well as

Fig. 5). Such evolution may be related to the observed decrease in modelled permafrost probability along the landform area (Fig. 5) and the climate evolution in this region: Rabatel et al. (2011) reported a warming trend of 0.19ºC decade$^{-1}$ for the 1958–2007 period in the Pascua-Lama area, 80 km north of Las Tetas. Monnier et al. (2014) also reported a trend of 0.17ºC decade$^{-1}$ for the 1974–2011 period in the Río Colorado area. Such evolution is hence reminiscent of reports of destabilization phenomena over rock glaciers in response to air and permafrost temperature increases (Roer et al., 2005 and 2008; Delaloye et al., 2010; Kellerer-Pirklbauer and Kaufmann, 2012).

### 5.3.3 Final diagnostics and future evolution of the landforms

According to the results and interpretations brought for the Navarro's eastern part and Presenteseracae, rock glaciers can develop at the expense of debris-covered glaciers, by an upward progression of their morphology and correlative widespread development of cohesive mass flow. These are true cases of debris-covered glaciers evolving in rock glaciers (see Introduction: type [iii]). At Presenteseracae, however, the flow does not appear as strikingly cohesive as in the Navarro's western unit, possibly due to the smaller size of the landform as well as a steeper slope that may favour subtle movement inconstancy. As these two landforms are located in favourable topoclimatic conditions, they should thus pursue their evolution towards rock glaciers. Monnier and Kinnard (2015) have nevertheless highlighted how the steep topography in front of Presenteseracae may represent an obstacle for its future development and sustainment. Despite the important insights brought by our study, it must be stressed out that the evolution of the internal structure in response to morphological and dynamical evolutions at the surface remains unknown; it would require decades of borehole and geophysical survey monitoring to properly assess this. However, and as suggested for Presenteseracae where the formation of distinct flow lobes was identified in the upper part, the transition may proceed by fragmentation of the glacier ice core and its mixing with debris and other types of ice (interstitial, intrusive) entrained from the surface. This is an alternative to the common and controverted model of the glacier ice-cored rock glacier where the evolution of the landform is controlled by the expansion and creep of a massive and continuous core of glacier ice (e.g., Potter, 1972; Whalley and Martin, 1992; Potter et al., 1998).

The Navarro's western unit and Las Tetas are more commonly known cases of assemblages that have formed and evolved in reaction to the superimposition/embedding of glaciers onto

or in the back of rock glaciers and their subsequent dynamical interactions (see Introduction: type [i]). In both cases, the progression of the rock glacier at the expense of the debris-covered glacier is rather limited (Navarro's western unit) or null (Las Tetas). At Navarro's western unit, the debris-covered glacier has displaced more slowly than the rock glacier. At Las Tetas, there is no major contrast in displacement speed between the debris-covered and the rock glacier. In both cases, the better depiction of flow patterns and their more extensive nature evoke a 'locomotive' role for the rock glacier. It is nevertheless difficult to assert whether the debris-covered glaciers are 'pushing away' the rock glaciers or if the latter are 'pulling' the former; both processes probably occur. The dynamical links between both units certainly constitutes a complex issue deserving more attention. The flow dynamics highlighted at Navarro supports the idea of a close relationship between the debris-covered and rock glacier, with many streamlines connecting both areas. At Las Tetas it is very conspicuous how during the studied period the debris-covered and the rock glacier changed from being dynamically coupled to essentially de-coupled after 1978. Furthermore, as these whole landforms continue to advance, the rock glaciers could plausibly become entirely isolated from their main debris source in the upper cirques while the increasingly warming conditions could cause the debris-covered glacier to become stagnant or disappear. Also, as the rock glaciers penetrate in areas with less favourable topoclimatic conditions, their future sustainment can be questioned.

## 6   Conclusion

We have used remote sensing techniques including imagery orthorectification, DEM comparisons, and cross-correlation image matching, in order to measure the morphological evolution, elevation changes, and horizontal displacements of three glacier–rock glacier transitional landforms in the central Andes of Chile over a human life-time scale. Our study highlights how, as climate changes and mountain landscapes and their related dynamics shift, the glacial and periglacial realms can strongly interact. The pluri-decadal landscape evolution at the three studied sites is noticeable but the modalities and significance vary between sites. Navarro and Las Tetas are composite landforms resulting from the alternation between glacier (re)advance and rock glacier development phases, and which currently exhibit either progression of the rock glacier morphology and associated cohesive mass flow and surface stabilization, or ice loss-related downwasting and surface destabilization features. Presenteseracae is a special case of small debris-covered glacier that has evolved into a rock

glacier during the last decades, with the rock glacier morphology having mostly developed ~15 years ago and with fast horizontal surface displacements (up to more than 3 m yr$^{-1}$). Topoclimatic conditions appear to have been determinants in the landforms' evolution and, by extrapolation, could thus be expected to exert an important control on the development and conservation of underground ice in high mountain catchments. From the latter point of view, our study stresses how spatial and dynamical interactions between glaciers and permafrost create composite landforms that may be more perennial than transitory: depending on the frequency of glacial–periglacial cycles, they participate in sustaining a hybrid cryospheric landscape, potentially more resilient against climate change. This conclusion is of societal importance considering the location of the studied landforms in semiarid areas and the warming and drying climate predicted for the coming decades (Bradley et al., 2006; Fuenzalida et al., 2006).

We have furthermore provided new insights into the glacier–rock glacier transformation problem. Most of the common and previous glacier–rock glacier evolution models depicted a 'continuum' process based on the preservation of an extensive core of buried glacier ice. On the contrary, our findings rather suggest that the transformation of a debris-covered glacier into a rock glacier may proceed from the upward progression of the rock glacier morphology at the expense of the debris-covered glacier, in association with an expanding cohesive mass flow regime and probable fragmentation of the debris-covered glacier into an ice–rock mixture with potential distinct flow lobes. The highlighted importance of topoclimatic conditions and corresponding morphologic evolutions also supports the inclusion of the permafrost criterion within the rock glacier definition.

## Acknowledgements

This study is part of the Project Fondecyt Regular No. 1130566 entitled: "Glacier-rock glacier transitions in shifting mountain landscapes: peculiar highlights from the central Andes of Chile." Fondecyt is the National Fund for Research and Technology in Chile. The authors want to thank Arzhan Surazakov who performed the image processing, and Valentin Brunat, who was involved in the software handling and related data management in the framework of a Master Thesis supported by the abovementioned project. The authors also thank Andreas Kääb and one anonymous referee for their important help in improving this manuscript.

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

| Site | Date | Horizontal ground RMSE (m) | | Number of GCPs |
|---|---|---|---|---|
| | | $x$ | $y$ | |
| Las Tetas | 1956 | 0.92 | 0.93 | 5 |
| | 1978 | 1.13 | 1.16 | 10 |
| Navarro valley | 1955 | 1.82 | 1.32 | 13 |
| | 2000 | 0.76 | 1.49 | 9 |

**Table 2.**

Uncertainty related to the measurement of annual rates of vertical and horizontal displacements
according to the period considered. The horizontal uncertainty takes into account both the ground
RMSE related to the orthorectification (Table 1) and the RMSE related to the co-registration step in
the cross-correlation image matching. The vertical uncertainty corresponds to the probability level
associated with the standard deviation ($\sigma$) of the vertical bias of the generated DEMs. In Navarro
valley, no reliable DEM could be generated from the 1955 aerial photos, which explains the absence
of data in the table for the 1955–2000 interval.

| Site | Period | Horizontal uncertainty ($m\ yr^{-1}$) | Vertical uncertainty ($m\ yr^{-1}$) | |
|---|---|---|---|---|
| | | | 1 $\sigma$ (66%) | 2 $\sigma$ (95%) |
| Las Tetas | 1956–1978 | 0.17 | 0.11 | 0.21 |
| | 1978–2012 | 0.14 | 0.10 | 0.21 |
| Navarro valley | 1955–2000 | 0.10 | — | — |
| | 2000–2014 | 0.15 | 0.05 | 0.10 |

**Table 3.**

Use of the maximum front displacement measured on orthophotos between times for defining the size of the search window involved in the CIAS algorithm. Further, the longest displacement computed by the program can be compared with the maximum front displacement, giving an indication of the method reliability.

| Landform | Time interval | Maximum front displacement measured | | Selected search window (pixels) | Longest displacement computed (m) |
|---|---|---|---|---|---|
| | | metres | pixels | | |
| Navarro | 1955−2000 | 30 | 60 | 150 | 41 |
| | 2000−2014 | <5 | ≥10 | 50 | 13 |
| Presenteseracae | 1955−2000 | 120 | 240 | 500 | 156 |
| | 2000−2014 | 25 | 50 | 100 | 32 |
| Las Tetas | 1956−1978 | 20 | 40 | 100 | 32 |
| | 1978−2012 | 35 | 70 | 150 | 48 |

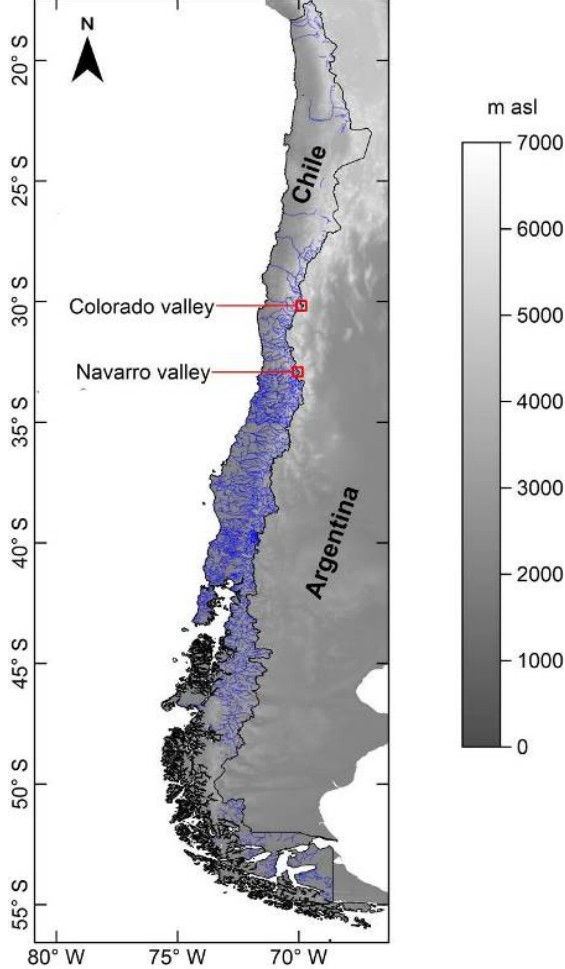

Figure 1. Location of the study sites. Drainage network, which reflects the variations of climatic–hydrologic conditions along the Chilean territory, is shown in blue.

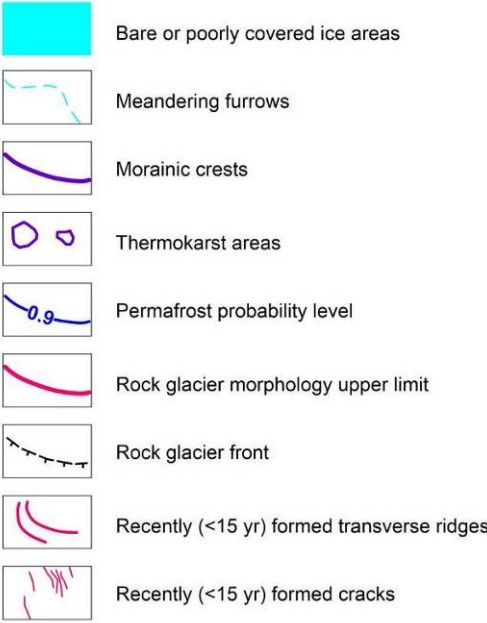

6   Figure 2. Geomorphological legend shared for all subsequent figures.

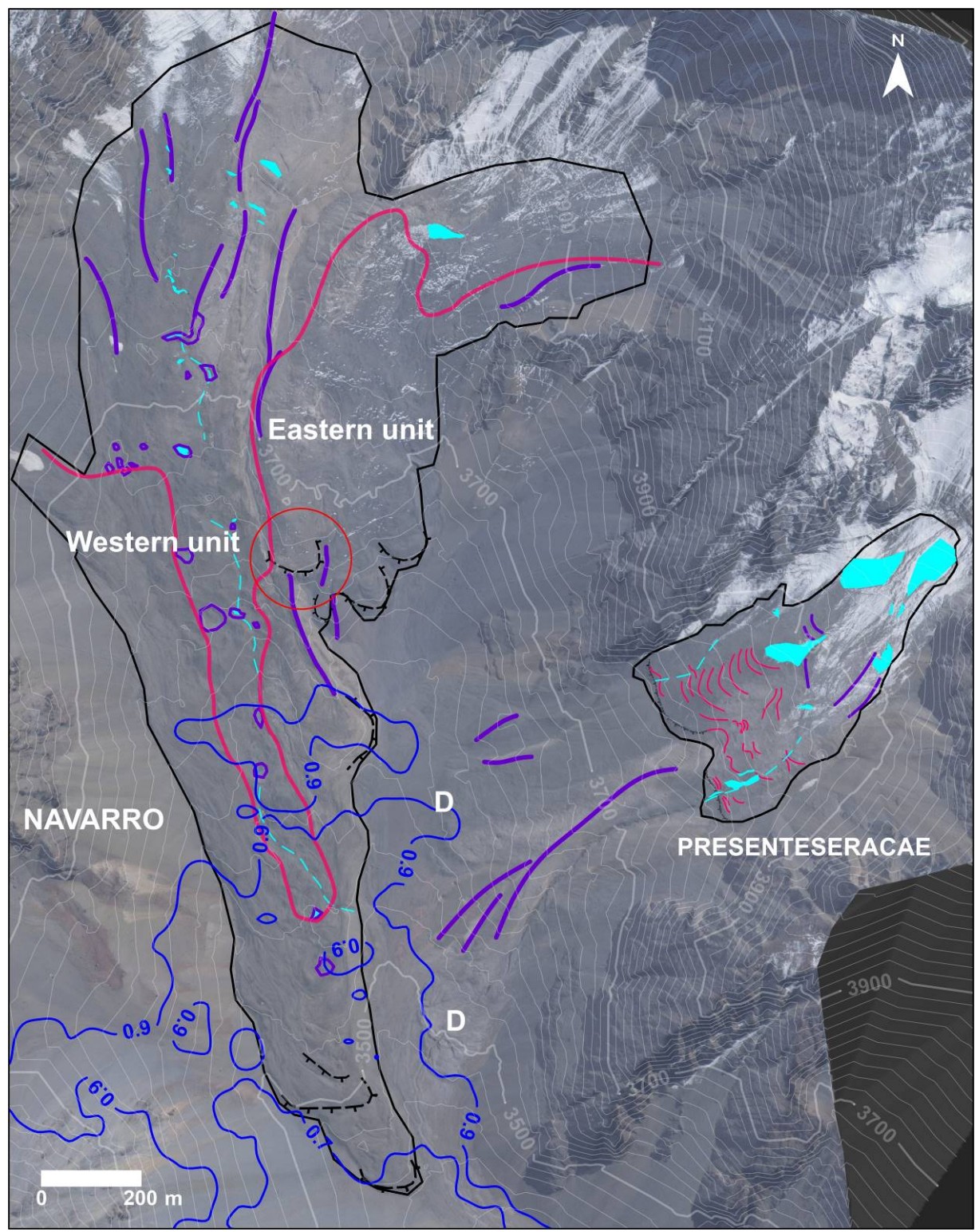

**Figure 3.** Map of the Navarro valley. See Fig. 2 for legend. The background of the map is the 2014
Geoeye image draped over the Geoeye DEM (see the Methods section). Elevation contours are derived
from the Geoeye DEM and contour interval is 10 m. The red circle indicates the location described in
the text where morainic crests and rock glacier lobes are superimposed. Note also the decayed (D)
rock glacier lobes in the area between Navarro and Presenteseracae.

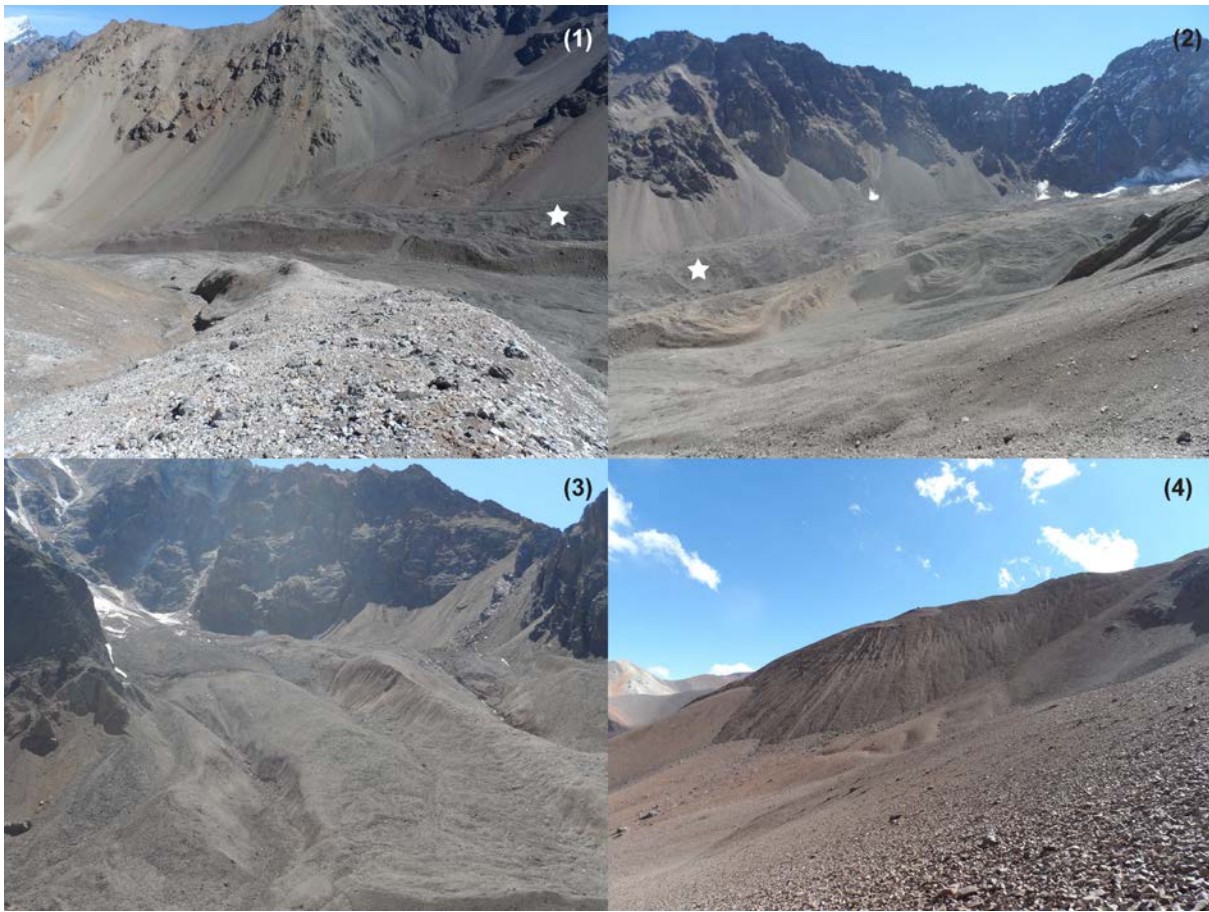

**Figure 4.** Photos of the lower (1) and upper part (2) of Navarro, seen from Presenteseracae,
Presenteseracae seen from Navarro (3), and the terminal part of Las Tetas (4) seen from its
northeastern surrounding area. The white star on pictures (1) and (2) indicates the main location of the
central depression and related thermokarst morphology on Navarro (see Fig. 3).

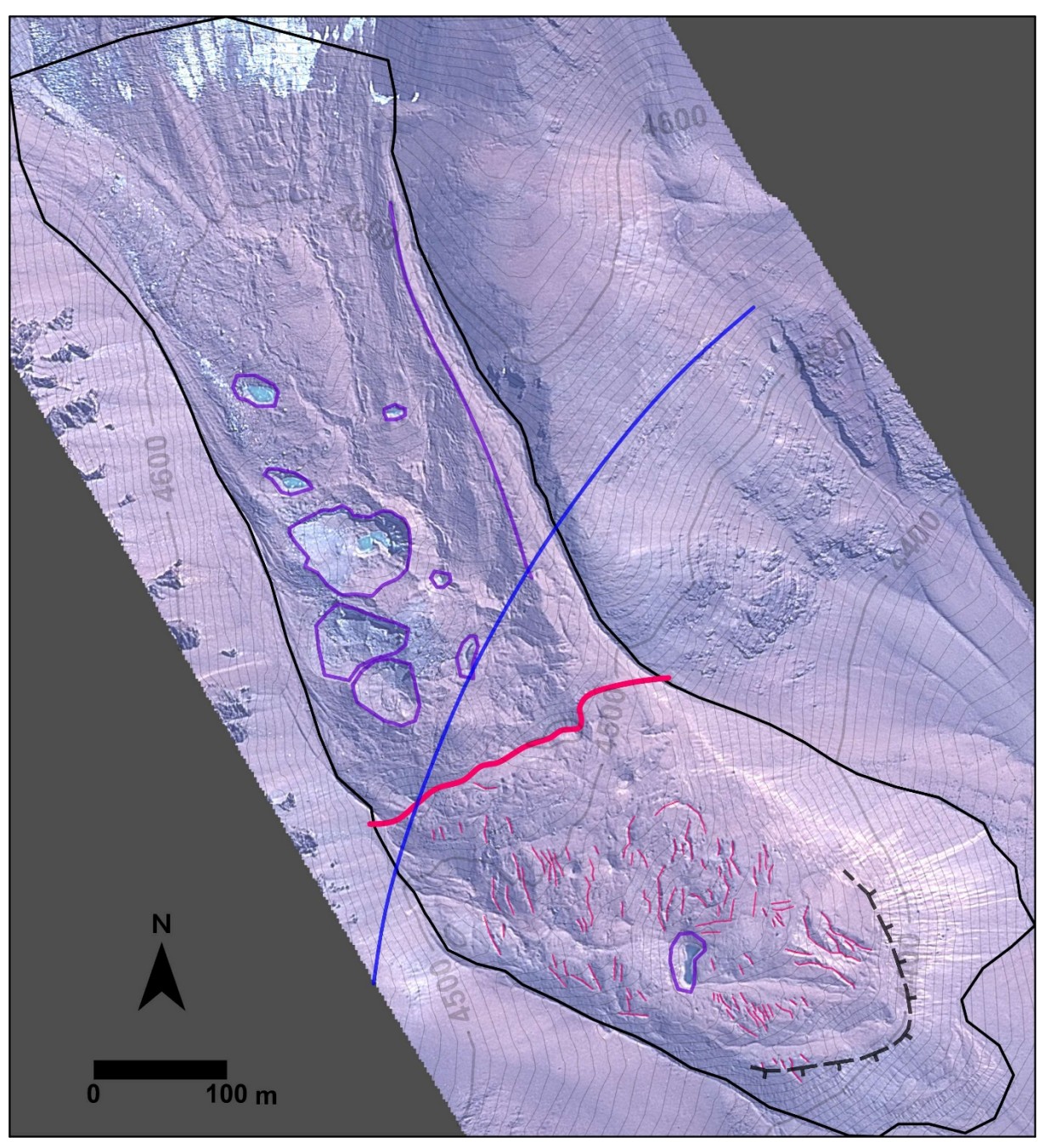

Figure 5. Map of the Las Tetas landform. See Fig. 2 for legend. The background of the map is the 2012 Geoeye image draped over the Geoeye DEM (see the Methods section).

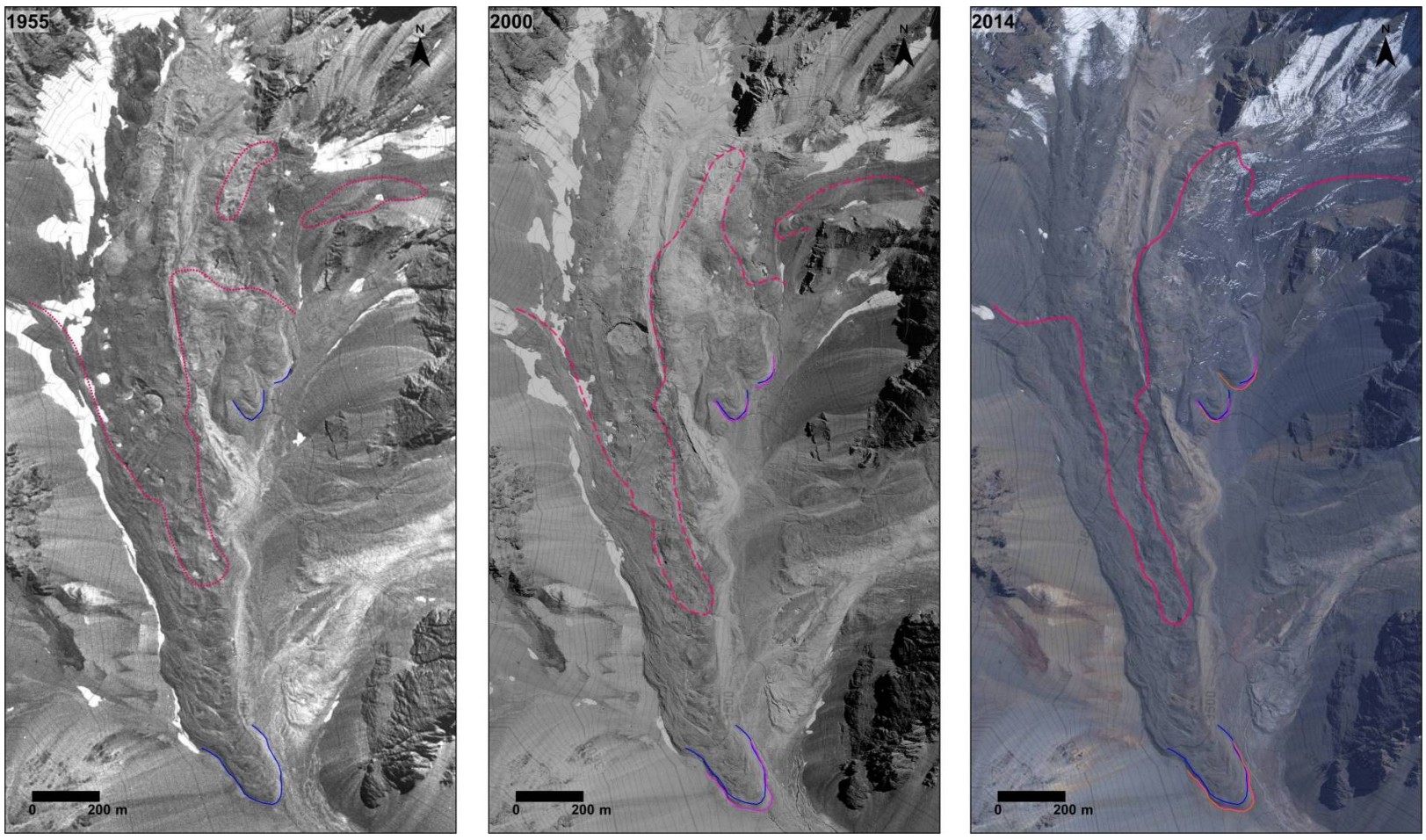

2 **Figure 6.** Sequence of orthophotos obtained for Navarro. When reliably identifiable the base of the front of the landform was indicated (blue, magenta, and
3 orange line in 1955, 2000, and 2014, respectively). At each date the boundary between debris-covered and rock glacier morphology is depicted with a red line
4 (dotted in 1955, dashed in 2000, continuous in 2014).

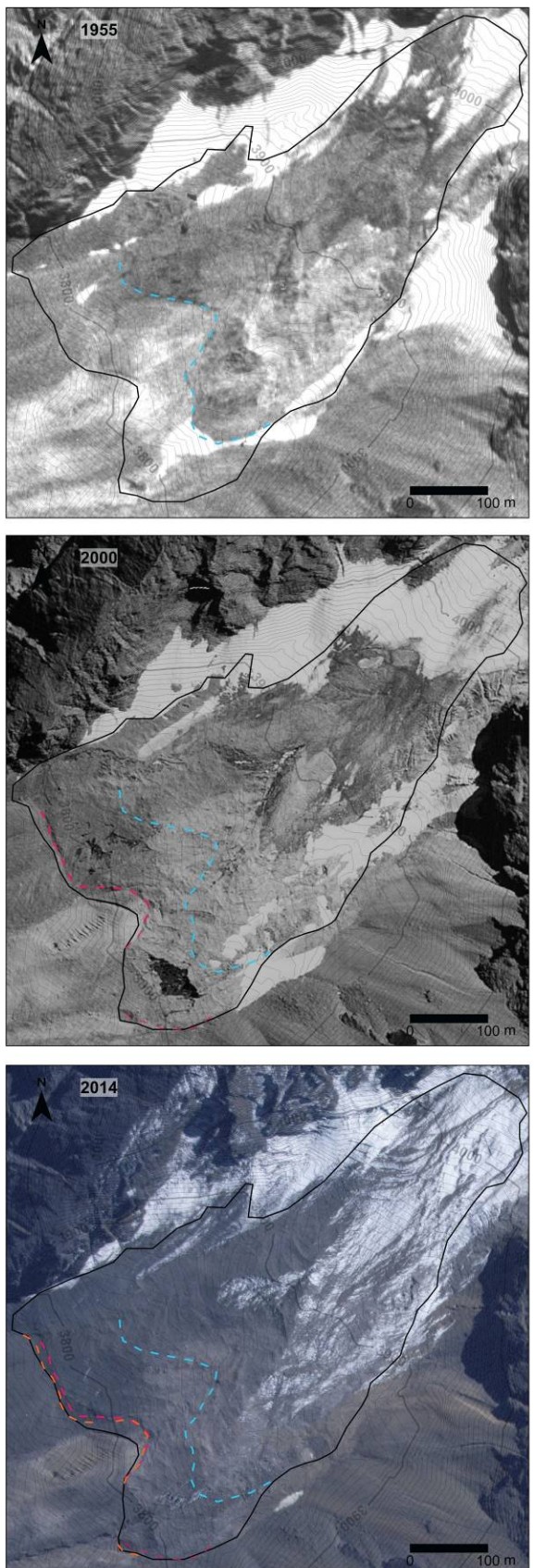

**Figure 7.** Sequence of orthophotos obtained for Presenteseracae. When reliably identifiable the base
of the front of the landform is indicated (blue, magenta, and orange dashed line in 1955, 2000, and
2014, respectively).

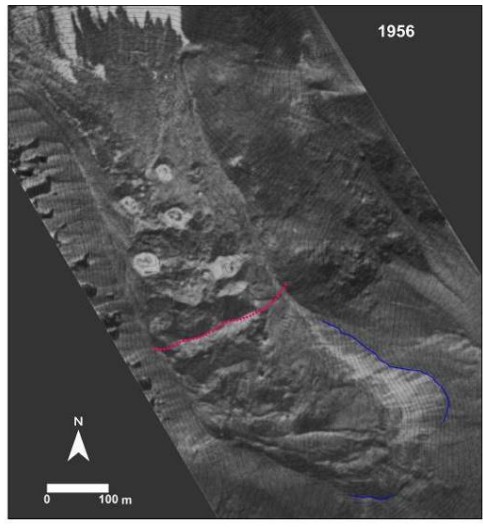

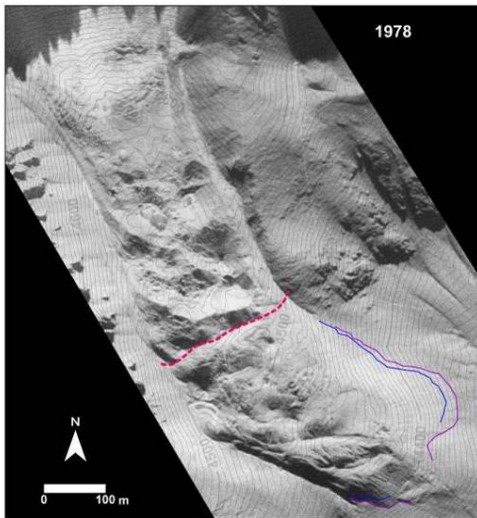

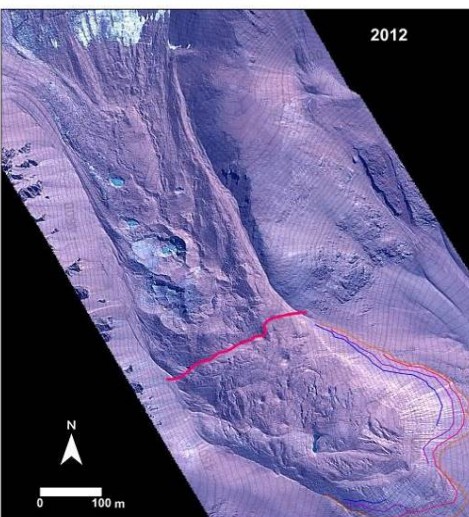

**Figure 8.** Sequence of orthophotos obtained for Las Tetas. When reliably identifiable the base of the
front of the landform is indicated (blue, purple, and orange line in 1956, 1978, and 2012, respectively);
the base of the front identified on a 2000 orthophoto not included in the study is also indicated
(magenta line). At each date the boundary between debris-covered and rock glacier morphology is
depicted with a red line (dotted in 1956, dashed in 1978, and continuous in 2012)

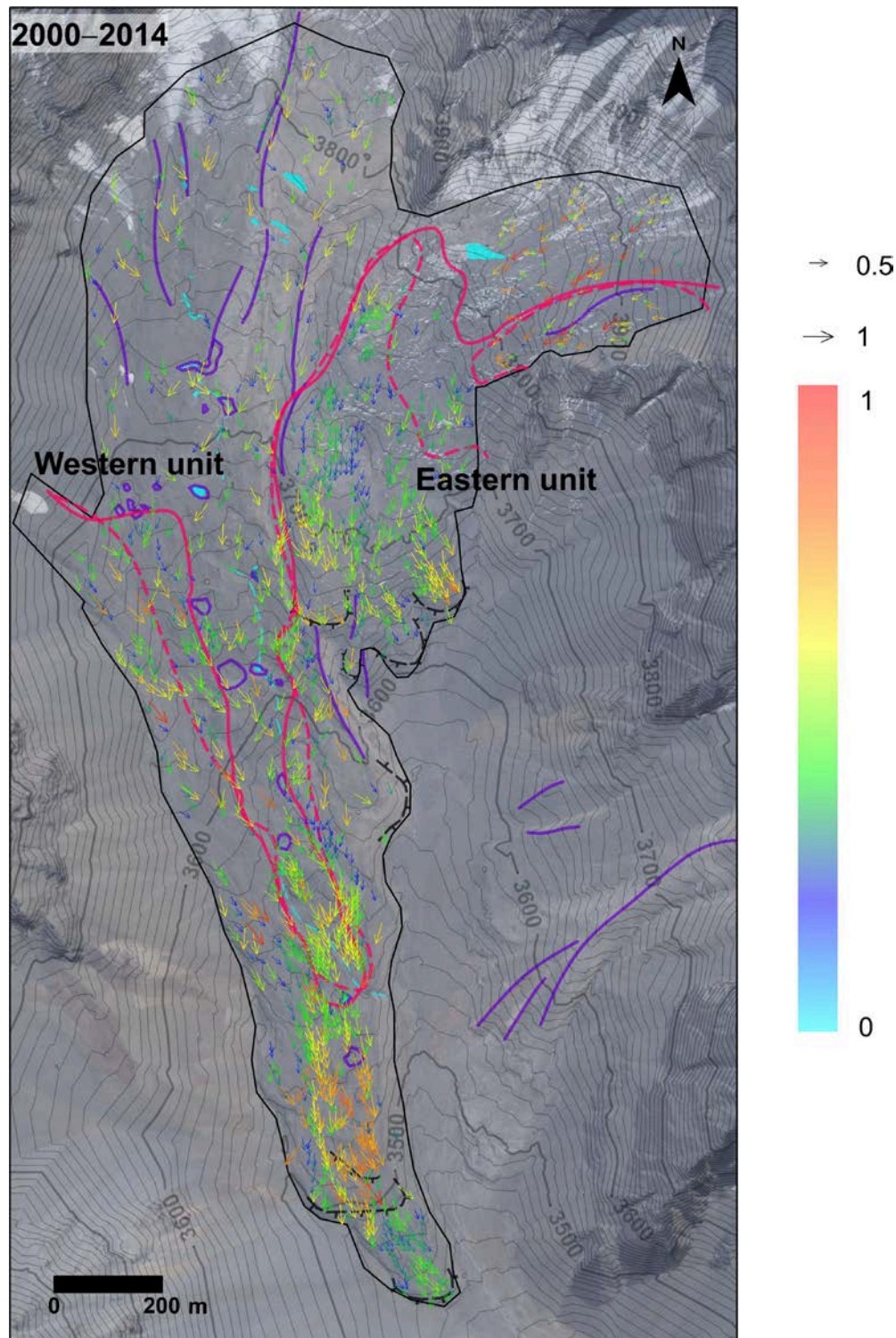

**Figure 9a.** Horizontal displacements (m yr$^{-1}$) at the surface of Navarro between 2000 and 2014. The
boundary between debris-covered and rock glacier morphology is depicted with a dashed red line in
2000 and with a continuous red line in 2014. Note that moraine crests, thermokarst depressions,
meandering furrows, and zones of bare ice in 2014 are indicated. Note that the orientation criteria for
horizontal displacement filtering were modified in the NE corner of the landform where the general
flow direction is radically different (dashed square area). The background of the map is the 2014
Geoeye image.

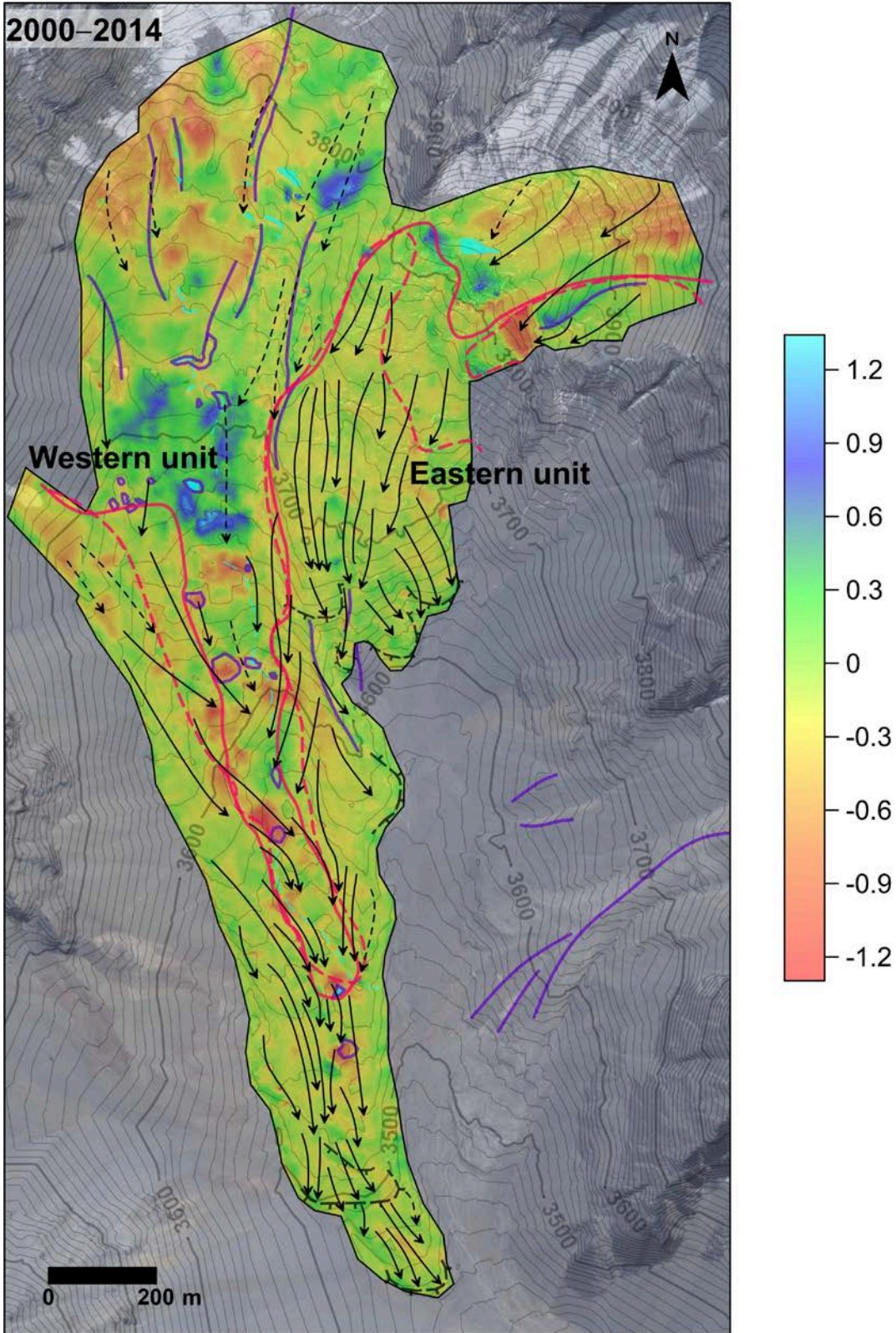

**Figure 9b.** Elevation changes and streamlines (derived from Fig. 9a) at the surface of Navarro between 2000 and 2014. The boundary between debris-covered and rock glacier morphology is depicted with a dashed red line in 2000 and with a continuous red line in 2014. Note that moraine crests, thermokarst depressions, meandering furrows, and zones of bare ice in 2014 are indicated. The background of the map is the 2014 Geoeye image.

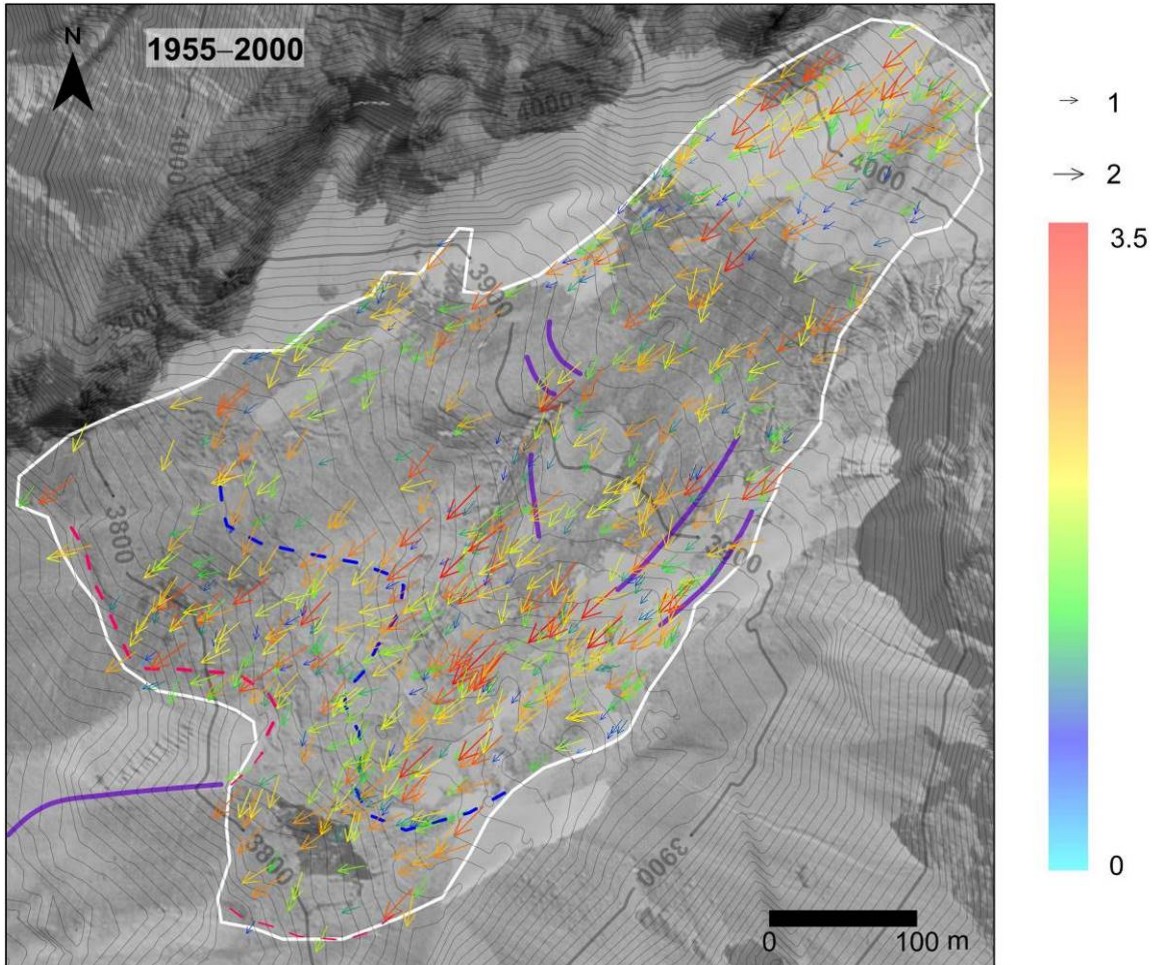

**Figure 10a.** Horizontal displacements (m yr$^{-1}$) at the surface of Presenteseracae between 1955 and
2000. The position of the base of the front at the two dates is indicated with dashed lines, as in Figure
7; push moraine ridges in the upper part are also indicated. The background of the map is the 2000
orthophoto.

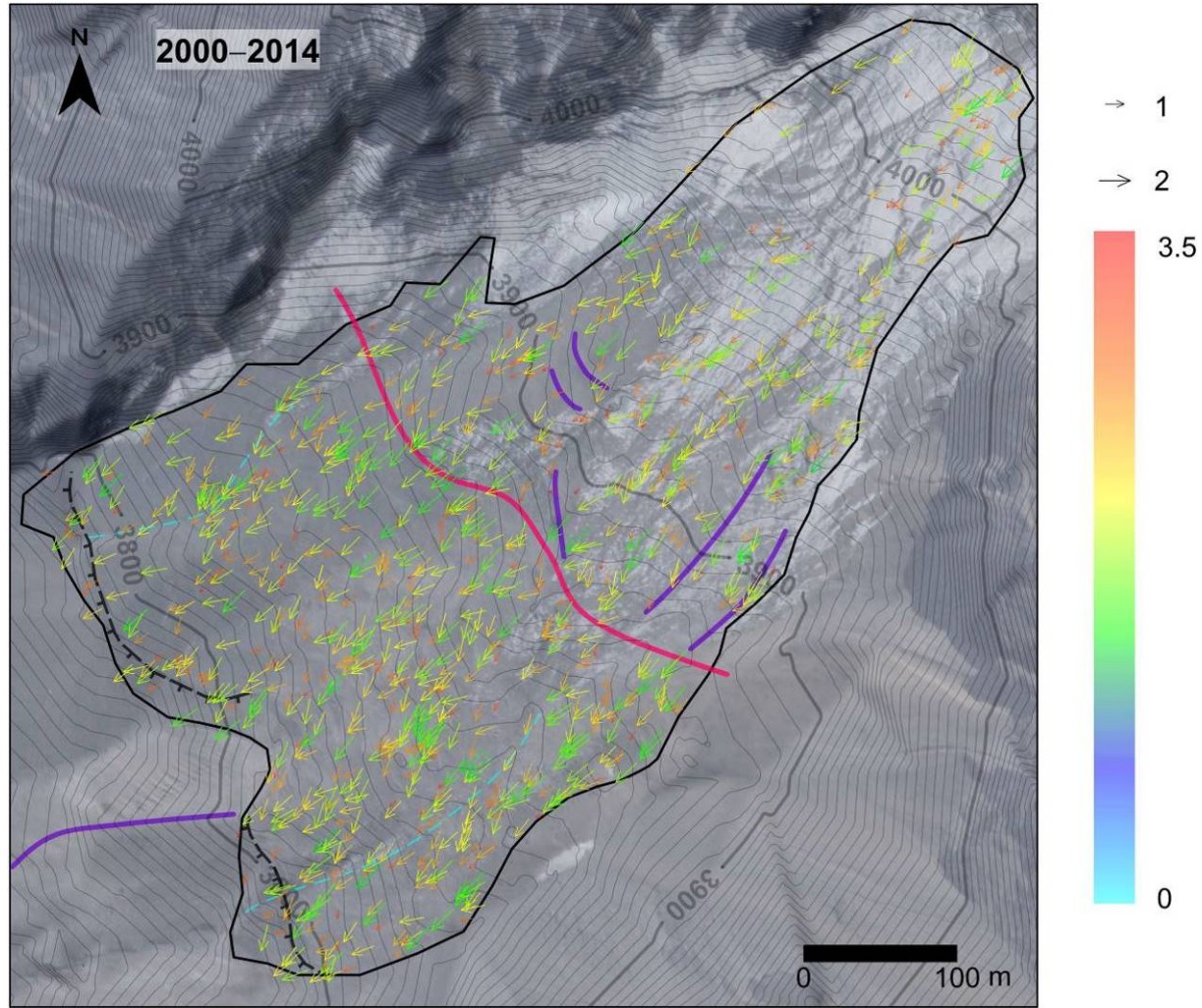

**Figure 10b.** Horizontal displacements (m yr$^{-1}$) at the surface of Presenteseracae between 2000 and
2014. The top of the front of the landform, the boundary between rock glacier and debris-covered
glacier features, and push moraine ridges in the upper part are indicated. The background of the map is
the 2014 Geoeye image.

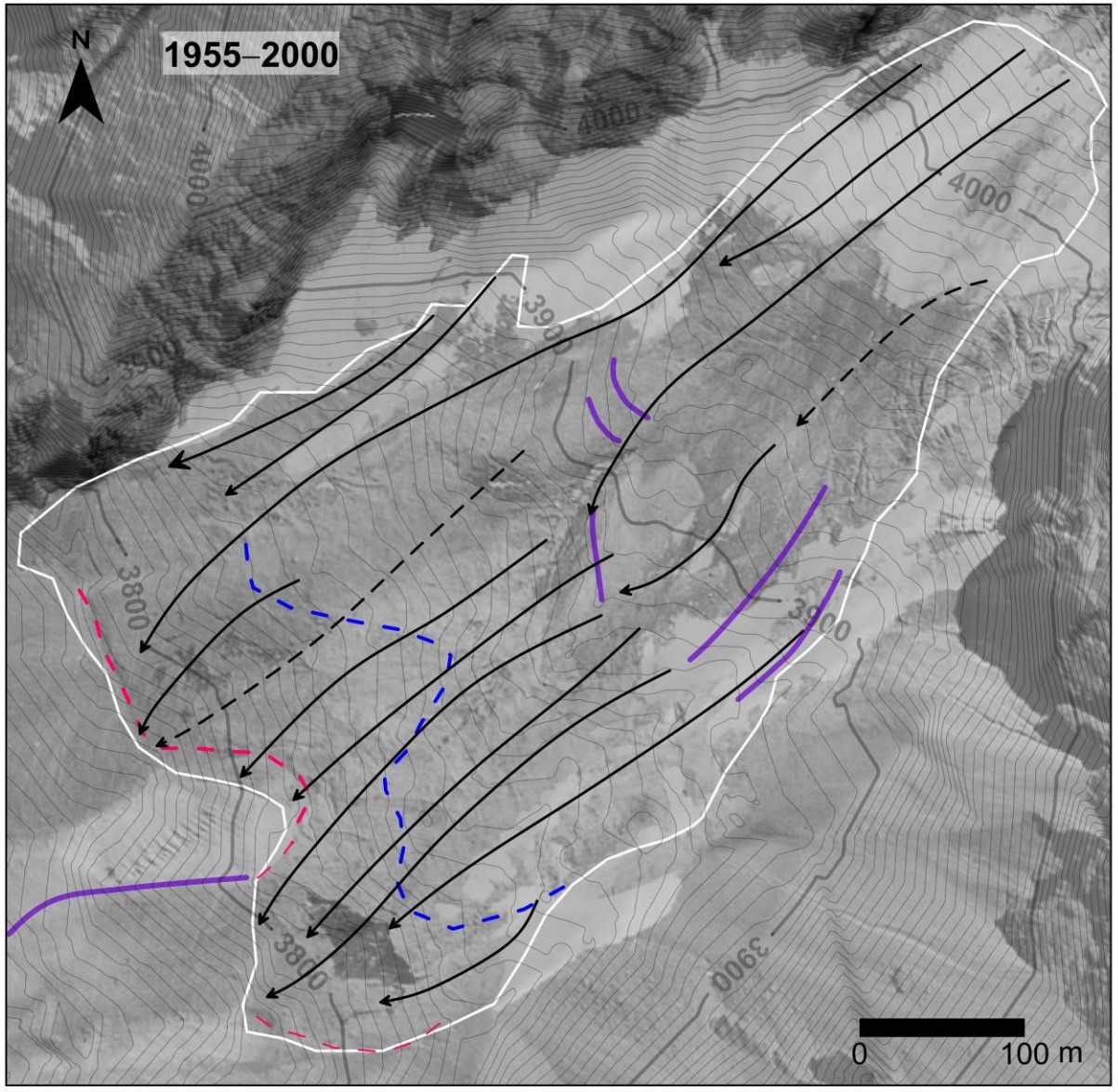

**Figure 11a.** Streamlines (derived from Fig. 10a) at the surface of Presenteseracae between 1955 and
2000. The position of the base of the front at the two dates is indicated with dashed lines, as in Figure
7; push moraine ridges in the upper part are also indicated. The background of the map is the 2000
orthophoto.

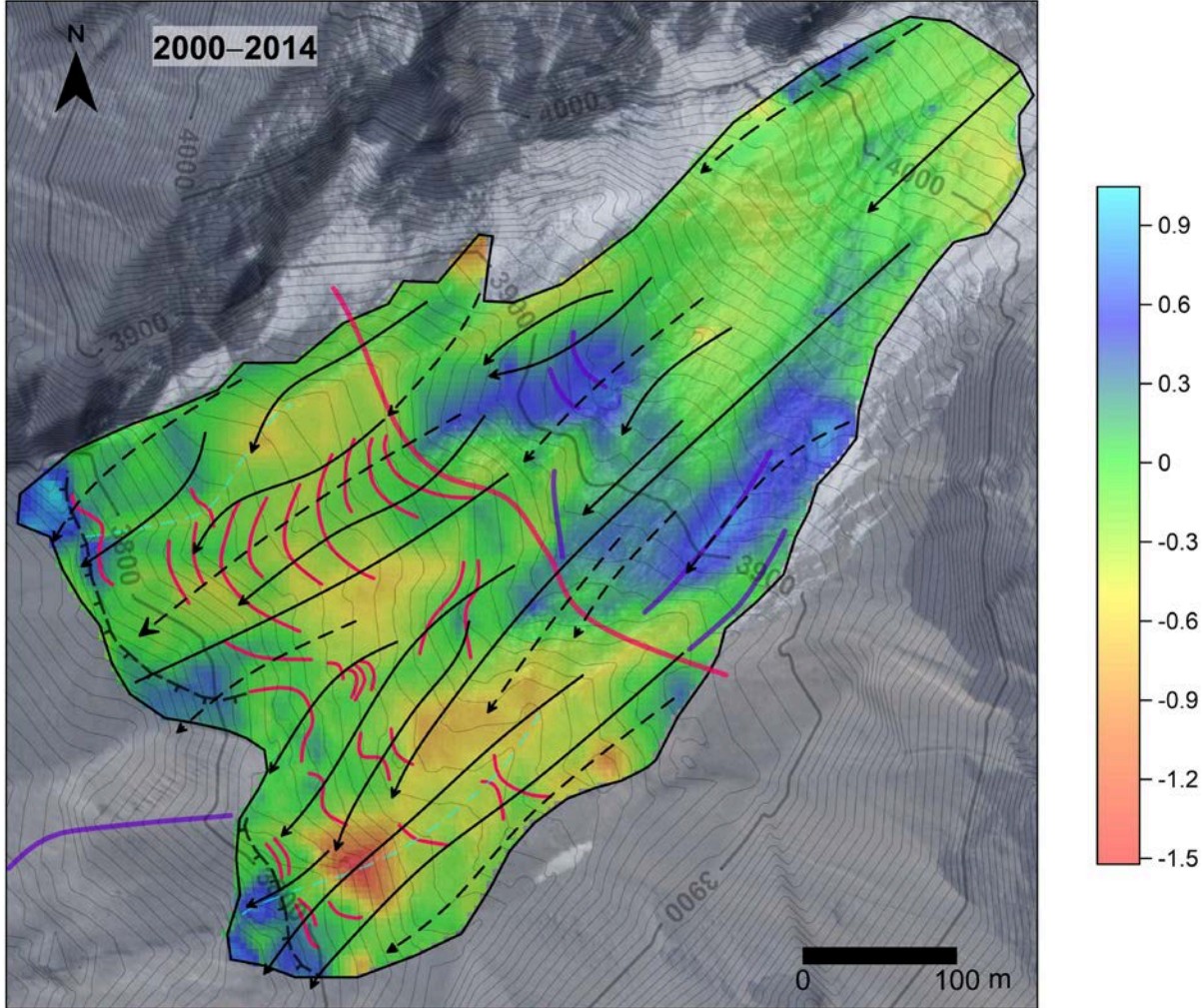

**Figure 11b.** Elevation changes (m yr$^{-1}$) and streamlines (derived from Fig. 10b) at the surface of Presenteseracae between 2000 and 2014. The top of the front of the landform, the boundary between rock glacier and debris-covered glacier features, cohesive flow-evocative ridges, and push moraine ridges in the upper part are indicated. The background of the map is the 2014 Geoeye image.

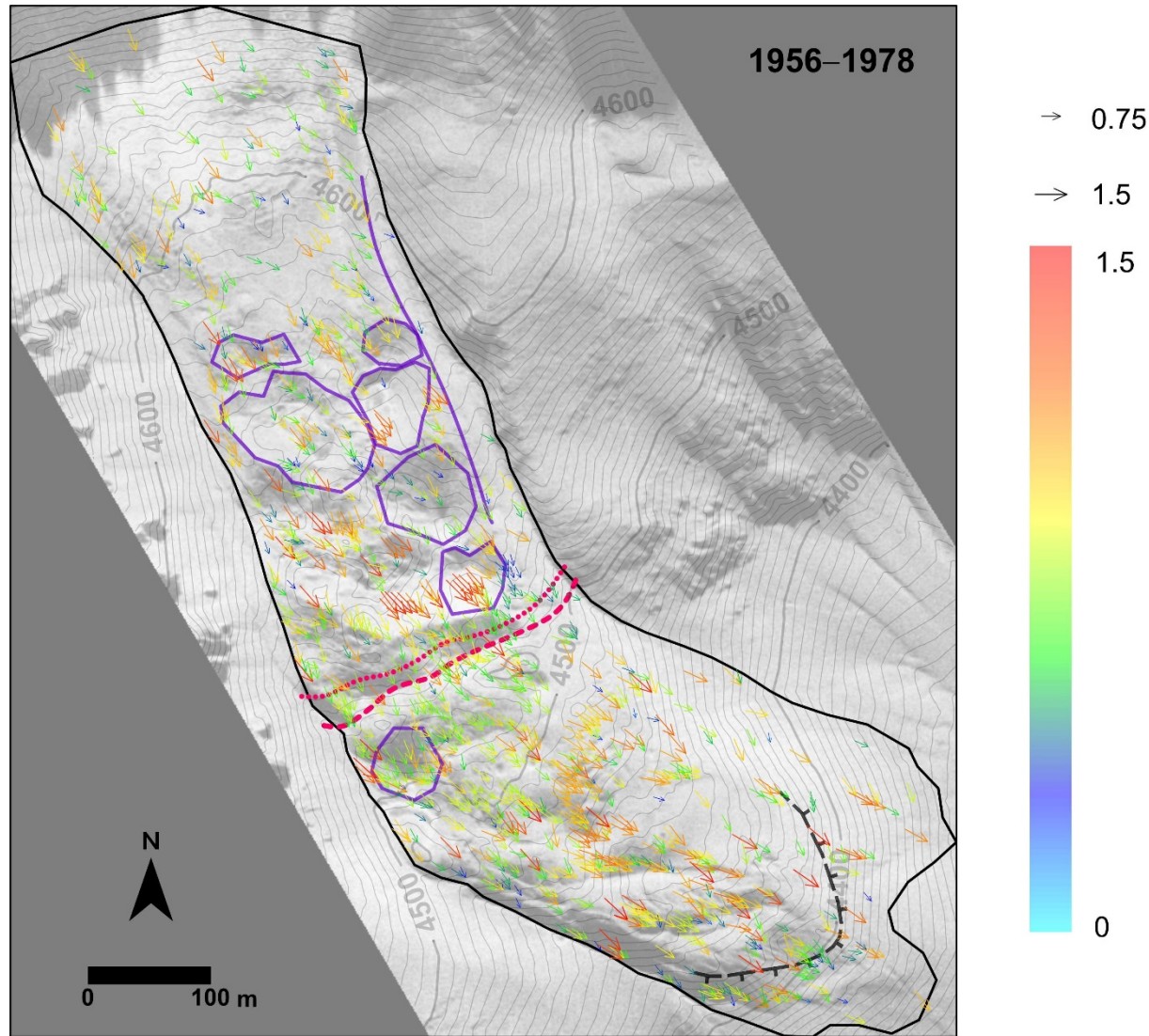

Figure 12a. Horizontal displacements (m yr$^{-1}$) at the surface of Las Tetas between 1956 and 1978. The boundary between debris-covered and rock glacier morphology is depicted with a dotted red line in 1956 and with a dashed red line in 1978. Note that thermokarst depressions in 1978 are indicated. The background of the map is the 1978 orthophoto.

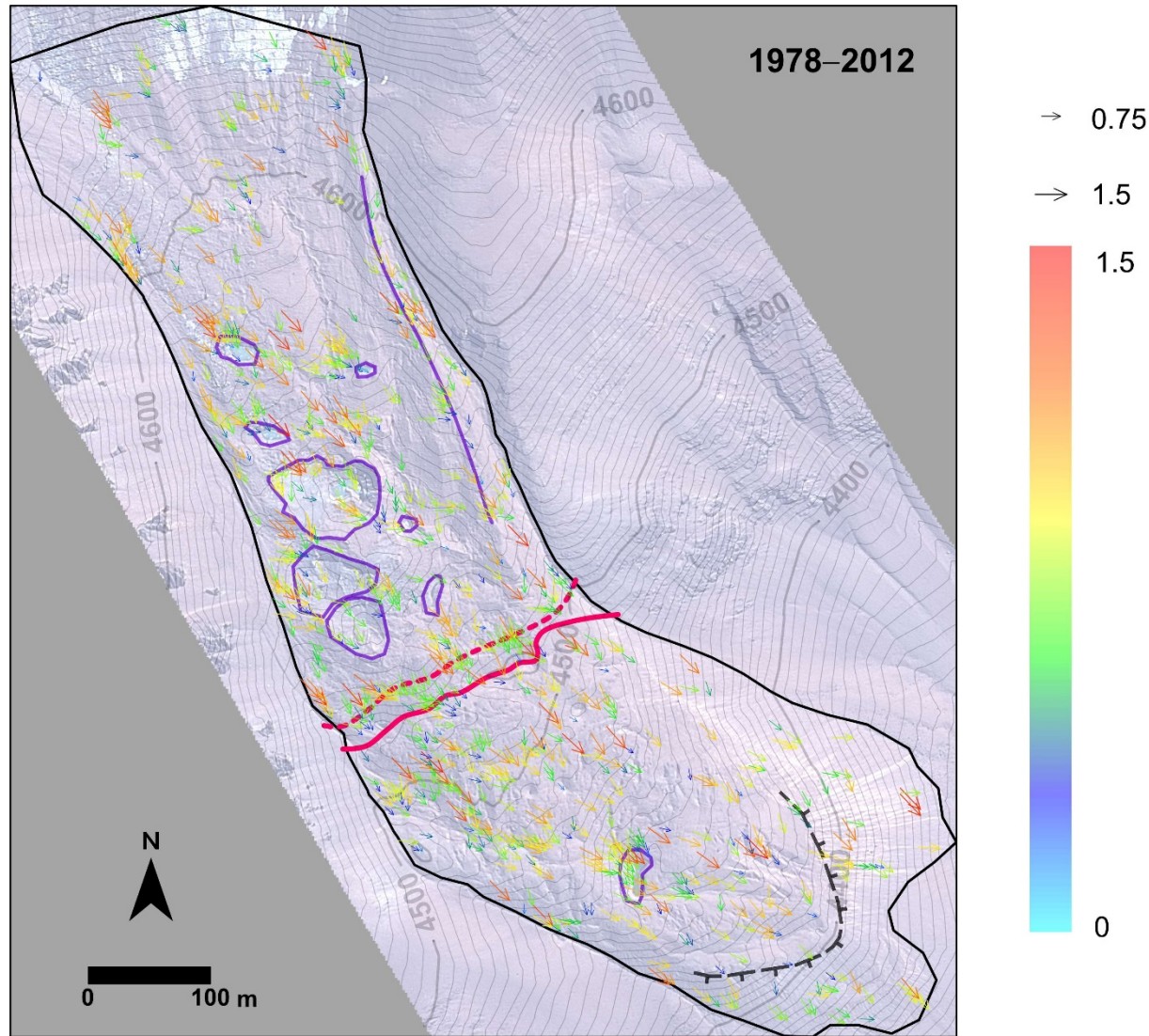

2 **Figure 12b.** Horizontal displacements (m yr$^{-1}$) at the surface of Las Tetas between 1978 and 2012.
3 The boundary between debris-covered and rock glacier morphology is depicted with a dashed red line
4 in 1978 and with a continuous red line in 2012. Note that thermokarst depressions in 2012 are
5 indicated. The background of the map is the 2012 Geoeye image.

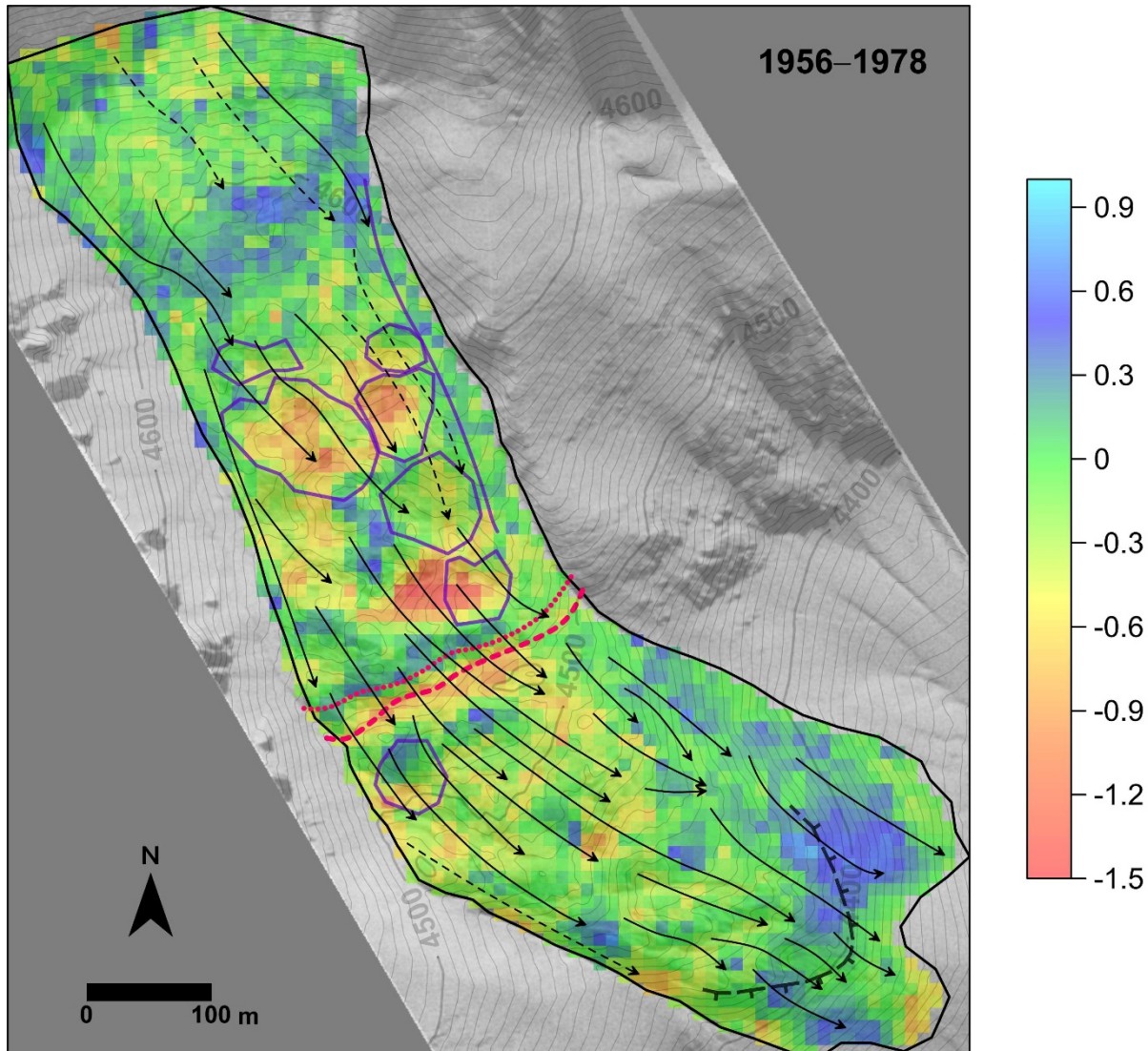

**Figure 13a.** Elevation changes (m yr$^{-1}$) and streamlines (derived from Fig. 12a) at the surface of Las Tetas between 1956 and 1978. The boundary between debris-covered and rock glacier morphology is depicted with a dotted red line in 1956 and with a dashed red line in 1978. Note that thermokarst depressions in 1978 are indicated. The background of the map is the 1978 orthophoto.

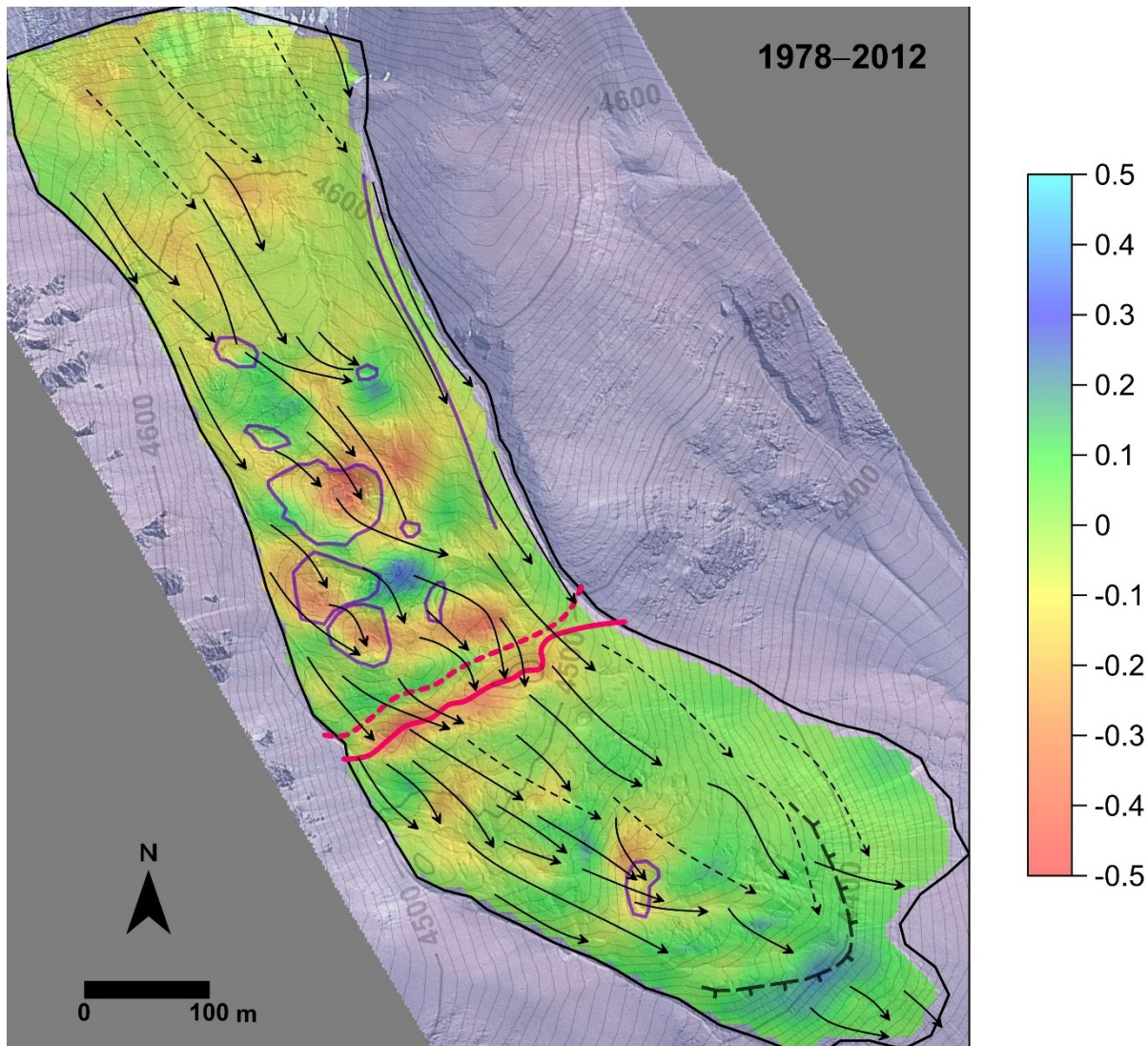

**Figure 13b.** Elevation changes (m yr$^{-1}$) and streamlines (derived from Fig. 12b) at the surface of Las Tetas between 1978 and 2012. The boundary between debris-covered and rock glacier morphology is depicted with a dashed red line in 1978 and with a continuous red line in 2012. Note that thermokarst depressions in 2012 are indicated. The background of the map is the 2012 Geoeye image.