# Peer review of "Pluri-decadal (1955–2014) evolution of glacier-rock glacier"

_Earth Surface Dynamics, 2016_

## Referee Comment (RC1) · A. Kääb (Referee) · 11 Apr 2016

This study discusses glacier-rockglacier transitions and changes over time of selected rockglaciers in the Andes of Chile. This is a very interesting and timely contribution on a less studied rockglacier area that can add to the understanding of rockglaciers and their climatic and geomorphological significance. The paper is well written and nicely illustrated. The paper could, though, benefit from a broader view and discussion of the changes seen.

I have however a major problem with the displacement measurements presented, which are a or even the core of the study. For me, most of the displacement vectors look like mis-matches, i.e. wrong measurements. In parts, especially on the glacier

parts, it is very hard for me to imagine that there are corresponding points that are preserved and can be matched over tens of years. Without checking the original images used, it is however not possible to me to judge this thoroughly. I offer to the authors to contact me directly for further details and solving my concerns. In sum, for me the quantitative results of the study are for now under a big question mark.

Further comments:

Page 7/line 14: Which software did you use for matching?

7/17: did you test your DEMs for lateral offsets and higher-order biases?

7/25: I don't understand fully: you used the airphoto DEMs for differencing, but they were too bad to use them for orthorectification?

8/1: better not to use 'vertical displacemenents' for elevation changes and thickness changes, as 'displacements' suggests that particles are moving verically, which is not the case for the type of DEM differences you observe.

8/12: You need to show and discuss the DEM differences on stable ground. This would be a good indicator of the uncertainty.

Show all images, not only for Presenteseracae Fig 8, so that the reader can judge by himself. Perhaps in an Appendix.

11/17, and 13/27: show and discuss stable ground displacements and DEM differences

17/16 and other places: you should discuss the processes potentially involved in the rockglacier/glacier changes observed more broadly and complete. For instance, what about potential changes in debris production, debris budget, debris evacuation, glacial transport, etc.

Andreas Kääb

---

## Referee Comment (RC2) · Anonymous Referee #2 · 29 Apr 2016

General comments

This is an interesting study on glacier – rock glacier relationship in arid mountain context. The scientific significance is quite good and it addresses relevant scientific questions. However, the overall importance of the study should be better highlighted. As presented in the introduction and in the discussion, the study focuses mainly on the study sites and lacks a broader view. The research question is not enough developed. Why is it important to make this study ? Also, the state of the art should be more developed. Some recent references are missing (eg. Dusik et al. 2014, ESPL; Bosson and Lambiel 2016, Frontiers in Earth Science).

Regarding the methods, a large part of the study relies on geomorphological charac-

teristics of the investigated landforms. Whereas the descriptions are generally clear and well presented, geomorphological mapping can be highly subjective. Then you must explain how the boundaries between rock glacier parts and debris-covered parts were defined. Which criteria were used to distinguish rock glacier morphologies from debris-covered ones ? Besides, I have a special concern with the horizontal velocities shown on the figures, for four reasons. 1) The vectors are so small – especially for site 3 – that the velocities are almost impossible to read. 2) The vectors appear quite noisy, with many different directions ; or they are so numerous that the information is lost. 3) A systematic error seems to be present for site one, with many vectors in the Southewest direction. 4) The velocities seem homogeneous at the landform scale, which means that no movement differences would exist between the upper and lower parts, or debris-covered glacier and rock glacier parts. This seems unrealistic. In the results section, please group the chapters Âń Rock glacier morphology Âż and Âń Thermokarst area Âż. This would avoid having chapters with only 1 line, like in page 12. You should also display all aerial images, for the reader to make his own analysis. We feel sometimes uncomfortable because it is not always possible to verify your interpretations.

My major concern is probably the discussion chapter, for several reasons. This chapter must be clearly improved, both the content and the structure. The conclusions appear quite speculative in some places; they are not always supported by data. The interpretation/discussion must be clearly improved, for the reader to be convinced. The processes involved in debris-covered derived rock glaciers must be better discussed. And this must be done without separating the three sites. In the current state, it is more an interpretation than a discussion chapter. You must better discuss your interpretations. For instance you mention compression and bulging in chapter 4.2.4. How do you reach these conclusions? This is unclear. Another example : You use the permafrost probabilities given by your modelling for the interpretation of spatial differences on case one. This seems simplistic : a model is not the truth, especially with permafrost spatial models. The reality is often much more complex at the site scale. Honestly I do not see

any clear difference between Navarro and Las Tetas regarding glacier – rock glacier interactions. How can you conclude that there was only a colliding between the glacier and the rock glacier in Las Tetas, and no superimposition by the glacier, as it is the case in Navarro ? What about the origin of both rock glaciers ? Are they glacier-derived rock glaciers, or talus rock glaciers overridden by a glacier ? You should discuss this. However, you must be very cautious with the interpretation on rock glacier origin, since you have no geophysical data to show the internal structure. The references are rather few. You must much more discuss your findings regarding previous studies.

Specific comments

p.1, l.18. Âń monitored Âż is not suitable here. Prefer Âń investigated Âż, Âń assessed Âż,. . .

p.2, l.9-18. Please add references.

p.2, l.11. The debris cover can be continuous.

p.2, l.14. Âń RG morphology is coherent, stable, . . .Âż. What is a coherent and stable morphology ? Please be more accurate.

Fig. 1 : Explain in the text the significance of the drainage network. In addition, it appears violet instead of blue on the map.

p.4, l.25 and Fig.3. It is unclear how the limit between the rock glacier parts and debris-covered parts were drawn. On Fig. 3 you should draw the limits of the Western and Eastern units.

Fig. 4 : Indicate the location of Las Tetas on Fig. 3. Indicate by arrows or signs the location of the central depression and of the thermokarst on the photos.

p.5, l.10-13. These lines are a bit fuzzy. The model used is only indicative and, in my opinion, cannot be used to explain in a so simple way the morphological differences encountered on the different parts of the landform. In addition, I observe that thermokarst

**ESurfD**

Interactive
comment

is also present were permafrost probability is higher than 0.95.

p.5, l.28-32 : In my opinion this is not really a rock glacier morphology, because the landform is very thin, especially the northern lobe and because the surface ridges are not classical compression ridges of block-ice mixture, but rather ridges due to subsurface ice flow.

p.5, l.29-30 : Âń Small morainic. . . Âż : I do not understand what you are talking on.

p. 6 l. 1 : remove Âń ice Âż l. 22 : 100 m high seems to be exaggerated l. 27 and subsequents : Âń aerial photos Âż is more appropriated than Âń air photo Âż

p.8, l.1. Prefer "altitudinal changes" to "vertical displacement". Valid for the rest of the text.

p.8, l.14. There is an issue with this interpretation of vertical changes. If a vertical change occurs locally, this means that an additional process to downslope movement occurs. Indeed, if only downslope movement occurs (for instance without ice melt, ice aggradation or compression), then the altitude at the same location will not change.

p.8, l.19. You must explain how this boundary was defined. Which criteria were used to distinguish rock glacier morphologies from debris-covered ones ? This is important since a large part of the study relies on this distinction.

p.9, l.3. As far as I know, it is not necessary that the two images must be taken from the same position. With remote sensed images it is even impossible.

p.9, l.15-20 and Tab. 3. It is unclear for me how the search window was defined : what is the relationship between columns 4 and 5 in Tab. 3 ? In addition, the maximum expected displacement can be well above the front displacement.

p.10, chap. 4.1.1. Without viewing the aerial images, it is impossible to see how you reached your conclusions about geomorphological changes. You give a change value of 75%, but how can you make a spatial quantification of the changes, knowing that dif-

ESurfD
ferences between rock glacier and debris-covered glacier morphologies are somewhat not well defined and probably quite subjective.

p.10, l.20. What is this Âń current morphology Âż ?

p.10, chap. 4.1.3 and Fig. 6&7. As already mentioned it is very difficult to read the velocities. In addition, the directions appear to be very noisy, and a systematic error seems to occur, as indicated by the general south-west direction.

p.10, l.27. What are exactly the displacement rates for rock glaciers ? The values given by Barsch and Haeberli may be true for a period running up to the early 2000's, but now several studies have shown that the velocities have strongly increased in the recent years, and that velocities above 1 m/y is very often the norm.

p.11, l.17. I do not see the heaving in the fronts and the margins.

p.11, l.18. See my comments concerning the Method (interpretation of vertical movements).

p.11, l.27-30. This evolution is very difficult to see on the images. Could you make a zoom on the frontal part ? In addition, try to better show the rock glacier morphology (in Fig. 4_3 we do not see the front).

p.12. chap. 4.2.3. Obviously there are no velocity differences between the debris-covered glacier part and the rock glacier part. Is this realistic ? Again, are you sure that there is no problem with the method ? l.17-18. I do not see the two flow lobes in the upper part.

p.12. chap. 4.2.4. Your conclusions regarding compression, bulging, expansion and ice melt are too direct and lack any demonstrations. How can you conclude to these processes ? This must be better explained and discussed. For instance at the front, the vertical gains may also be attributed to the rock glacier advance and not by compression : in 2000 the rock glacier front was ( ?, you can verify) located upside the current position, so the blue colours here would just indicate the downslope progression of the

**ESurfD**
front.

p.13, chap. 4.3.3. The vectors are unreadable

p.15, l.29-32. I don't understand this sentence.

---

## Author Comment (AC1) · 26 Jun 2016

We sincerely acknowledge the two referees for the meticulous examination they did of our study and the valuable critics, remarks, and advices they provided.

Both referee highlighted issues related to the use of cross-correlation image matching, i.e., the presence of mismatches in the presented horizontal displacements, and inefficient graphical rendering. They were correct and we spent much time addressing this issue. We identified two problems: (i) an incorrect importation of the vector directions in the mapping software, and (ii) an unappropriated definition of the filtering criteria of the measurements. We have now resolved these problems. In particular, we used stricter criteria to filter the measurements performed (see below). In addition, we

interpreted the main stream lines from the displacement vector maps produced and presented them in separate (sub)figures, in order to help the interpretation.

Consequently, most parts of the Results and Discussion sections as well as several figures have been reworked and now differ significantly from the initial version of the manuscript, in content and in structure. Especially, we had to renounce to the use of the 1955 aerial image in the case of the Navarro landform, and of the 2000 aerial photo for the Las Tetas landform; the former produced two much noisy and mismatched data while the latter gave coherent but very sparse and therefore unsatisfactory results. Consequently, we now present displacement measurements for the 2000-2014 period at Navarro, for the 1955-2000 and 2000-2014 intervals at Presenteseracae, and for the 1956-1978 and 1978-2012 intervals at Las Tetas. Also, for each site we included new figures with the original images. Finally, we did our best to enhance the graphical design and rendering of the displacement and altitudinal change figures.

In the detailed responses to referees' comments we sometimes refer directly to the current General comment.

Author's methodological change in manuscript: see supplement file. See also all the corresponding figures.

Please also note the supplement to this comment:
http://www.earth-surf-dynam-discuss.net/esurf-2016-16/esurf-2016-16-AC1-supplement.pdf
* * *
[Figure]

**Supplement:**

**Author's change in manuscript**

Regarding the filtering criteria applied to horizontal displacements, see the Methods section (p. 11, l. 8-26 in the new version): "The subsequent filtering procedure excluded points that did not meet the following conditions: (i) the displacement magnitude must be higher than the RMSE generated by the orthorectification and co-registration steps (Table 2); (ii) the azimuthal deviation from the general landform flow direction must be less than 40º; (iii) the maximum cross-correlation coefficient in the search template around the point ($r_{max}$) must be higher than $\bar{r}_{max}$, the latter $\bar{r}_{max}$ being the maximum cross-correlation coefficient's average value for all measured points; (iv) the average cross-correlation coefficient in the search template must be less than $\bar{r}_{avg} + 1\sigma$, the latter $\bar{r}_{avg}$ and $\sigma$ being the average cross-correlation coefficient's average and standard deviation values, respectively, for all measured points. The use of the latter criterion aimed at removing noise data from the results, as suggested for example in the online CIAS recommendation. Eventually, remaining mismatches were manually removed after the filtering. The total displacements measured by the program were converted in annual displacement rates, and the displacement vectors were mapped."

---

## Author Comment (AC2) · 26 Jun 2016

**Responses to Referee 1 (Andreas Kääb)**

**Comment from referee, #1**
This study discusses glacier-rockglacier transitions and changes over time of selected rockglaciers in the Andes of Chile. This is a very interesting and timely contribution on a less studied rockglacier area that can add to the understanding of rockglaciers and their climatic and geomorphological significance. The paper is well written and nicely illustrated. The paper could, though, benefit from a broader view and discussion of the changes seen.

**Author's response**
As also suggested by the second referee, we have widened the view, scope, and discussion of our study. The overall significance of the study is better highlighted by covering the following key aspects: the spatial and dynamical interactions between glaciers and permafrost, the in vivo observation of rock glacier development, and how these composite landforms participate in sustaining a hybrid cryospheric landscape in which glacial and periglacial realms spatially and dynamically interact in response to climate change. The latter aspect is of particular importance in semiarid areas as the one studied here. We also insist much more on the importance of the rock glacier morphology, i.e. how it is likely to influence the surface energy balance, the subsurface heat transfers and resulting landform dynamics.

**Author's change in manuscript**
See in particular the whole introduction, discussion, and conclusion.

**Comment from referee, #2**
I have however a major problem with the displacement measurements presented, which are a, or even the core of the study. For me, most of the displacement vectors look like mismatches, i.e. wrong measurements. In parts, especially on the glacier parts, it is very hard for me to imagine that there are corresponding points that are preserved and can be matched over tens of years. Without checking the original images used, it is however not possible to me to judge this thoroughly. I offer to the authors to contact me directly for further details and solving my concerns. In sum, for me the quantitative results of the study are for now under a big question mark.

**Author's response**
See our General comment. The analysis has been re done completely and new figures produced.

**Comment from referee, #3**
Page 7/line 14: Which software did you use for matching?

**Author's response**
Erdas was used.

**Comment from referee, #4**
7/17: did you test your DEMs for lateral offsets and higher-order biases?

**Author's response**
The aerial photo DEMs were co-registered to the Geoeye DEM using common GCPs during DEM processing. The bias of the generated DEMs was tested over stable areas outside the

landforms' area, as mentioned in the Method section. Nevertheless, we did not analyse biases dependent on the elevation or geometry of the data acquisition (Nuth and Kääb, 2012).

**Author's change in manuscript**
p. 8, l.22-24 (in the new version): 'The same processing scheme was followed for the aerial photo stereo pairs using control points visible both on the Geoeye image and the aerial photo stereo pairs.'
* * *
**Comment from referee, #5**
7/25: I don't understand fully: you used the airphoto DEMs for differencing, but they were too bad to use them for orthorectification?

**Author's response**
The phrase was incorrectly formulated. We rephrased.

**Author's change in manuscript**
p. 9, l.1-3: "The Geoeye images were pansharpened and orthorectified using the Geoeye DEM. The aerial photos were then orthorectified using the corresponding DEMs, except when no reliable DEM could be obtained (as for 1955 at Navarro); in that case the Geoeye DEM was used."
* * *
**Comment from referee, #6**
8/1: better not to use 'vertical displacements' for elevation changes and thickness changes, as 'displacements' suggests that particles are moving vertically, which is not the case for the type of DEM differences you observe.

**Author's response**
We have replaced 'vertical displacements' by 'elevation changes' throughout the manuscript.
* * *
**Comment from referee, #7**
8/12: You need to show and discuss the DEM differences on stable ground. This would be a good indicator of the uncertainty.

**Author's response**
See (p. 8, l. 24-26; this was already mentioned in the first version): "The vertical bias of the aerial photo DEMs was calculated by comparison with the Geoeye DEMs over flat and stable areas outside the landform studied and was removed from the subsequent calculations (see below)." The biases appear in Table 2. They were also taken into account for the interpretation of elevation changes (p. 9, l. 18-21).
* * *
**Comment from referee, #8**
Show all images, not only for Presenteseracae Fig 8, so that the reader can judge by himself. Perhaps in an Appendix.

**Author's response**

We have included two additional figures, for Presenteseracae and Las Tetas, respectively. All original images are now visible in the article.

**Author's change in manuscript**
Fig. 6.
* * *
**Comment from referee, #9**
11/17, and 13/27: show and discuss stable ground displacements and DEM differences.

**Author's response**
See our response to Comment #7.
* * *
**Comment from referee, #10**
17/16 and other places: you should discuss the processes potentially involved in the rockglacier/glacier changes observed more broadly and complete. For instance, what about potential changes in debris production, debris budget, debris evacuation, glacial transport, etc.

**Author's response**
Absolutely. We developed a wider discussion in the new version of the manuscript. The composition of the discussion is now as such: 5.1. Initial landform development. 5.2. Differences between debris-covered and rock glacier areas. 5.3. Current evolution and its significance. 5.3.1. Landscape evolution. 5.3.2. Dynamical evolution. 5.3.3. Final diagnostics and future evolution of the landforms.

**Author's change in manuscript**
See the whole new discussion section in the text

**References (not cited in the manuscript)**
Nuth, C., Kääb, A.: Co-registration and bias corrections of satellite elevation data sets for quantifying glacier thickness changes. The Cryosphere, 5, 271–290. http://www.the-cryosphere.net/5/271/2011/tc-5-271-2011.pdf

---

## Author Comment (AC3) · 26 Jun 2016

**Responses to Referee 2**

**Comment from referee, #1**

This is an interesting study on glacier – rock glacier relationship in arid mountain context. The scientific significance is quite good and it addresses relevant scientific questions. However, the overall importance of the study should be better highlighted. As presented in the introduction and in the discussion, the study focuses mainly on the study sites and lacks a broader view. The research question is not enough developed. Why is it important to make this study?

**Author's response**

We acknowledge that the overall importance and insights given by our study were insufficiently exposed in the first version of our manuscript. We have made several additions to the new version in order to compensate this. As answered to Comment #1 from Referee 1: "The overall significance of the study is better highlighted by covering the following key aspects: the spatial and dynamical interactions between glaciers and permafrost, the in vivo observation of rock glacier development, and how these composite landforms participate in sustaining a hybrid cryospheric landscape in which glacial and periglacial realms spatially and dynamically interact in response to climate change. The latter aspect is of particular importance in semiarid areas as the one studied here. We also insist much more on the importance of the rock glacier morphology, i.e. how it is likely to influence the surface energy balance, the subsurface heat transfers and resulting landform dynamics."

**Author's change in manuscript**

See the whole introduction, discussion, and conclusion.

**Comment from referee #2**

Also, the state of the art should be more developed.

**Author's response**

In the introduction, we added various references and a whole paragraph oriented towards a better definition of the differences between debris-covered and rock glacier morphology in such transitional landforms (see also our answer to Comment #4).

**Author's change in manuscript**

See the whole introduction.

**Comment from referee, # 3**

Some recent references are missing (eg. Dusik et al. 2014, ESPL; Bosson and Lambiel 2016, Frontiers in Earth Science).

**Author's response**

Thank you for noticing and suggesting. These two references were included.

**Author's change in manuscript**

See Introduction.
* * *
**Comment from referee, # 4**

Regarding the methods, a large part of the study relies on geomorphological characteristics of the investigated landforms. Whereas the descriptions are generally clear and well presented, geomorphological mapping can be highly subjective. Then you must explain how the boundaries between rock glacier parts and debris-covered parts were defined. Which criteria were used to distinguish rock glacier morphologies from debris-covered ones?

**Author's response**

We agree that a big challenge with geomorphological interpretations is the use of a solid and objective set of criteria in order to compensate for the subjective aspect of the exercise. We agree that the first version of our manuscript was lacking a more exhaustive presentation of the morphological differences between debris-covered and rock glacier and of the importance of such differences.

**Author's change in manuscript**

See the introduction (p. 1-2): "Rock glaciers and debris-covered glaciers exhibit distinct morphologies that are of critical importance in the surface energy balance and subsurface heat transfer. On their surface, rock glaciers exhibit "the whole spectrum of forms created by cohesive flows" (Barsch, 1992, p. 176) of "lava-stream-like (…) viscous material" (Haeberli, 1985, p. 92). These features vary upon the case and study area; according to our field surveys in the Andes, they can be grouped in three main types: small-scale (<1 m high) ripples or undulations resulting from deformations in the active debris layer moving together with the underlying perennially-frozen core; medium-scale (1−5 m high) ridge-and-furrow assemblages resulting from the compression in the whole ice−debris mixture; and large scale (5−20 m thick and >100 m long) superimposed flow lobes upon which the first two types may naturally appear. Hereafter, we will simply refer to these features as 'cohesive flow-evocative features'. Contrarily, debris-covered glaciers are characterized by a chaotic distribution of features evocating surface instability such as hummocks, collapses, crevasses, meandering furrows, and thermokarst depressions and pounds. As a consequence, on rock glaciers the large- and fine-scale surface topography is rather smooth and convex, whereas on debris-covered glaciers it is rather rough and concave. Another morphological difference is the presence of ice visible from the surface: whereas ice is generally invisible from the surface of rock glaciers, it is frequently exposed on debris-covered glaciers due to the discontinuity of the debris cover or the occurrence of the aforementioned morphological features. Finally, and correlatively, over pluri-annual to pluri-decadal periods the morphology of well-developed rock glaciers is stable (beside cases of climate warming-related destabilizations, the geometry of surface features evolves but their overall pattern remains the same) while debris-covered glacier morphology is characterized by instability (surface features rapidly appear and disappear)."
* * *
**Comment from referee, # 5**

My major concern is probably the discussion chapter, for several reasons. This chapter must be clearly improved, both the content and the structure. The conclusions appear

quite speculative in some places; they are not always supported by data. The interpretation/ discussion must be clearly improved, for the reader to be convinced. The processes involved in debris-covered derived rock glaciers must be better discussed. And this must be done without separating the three sites. In the current state, it is more an interpretation than a discussion chapter. You must better discuss your interpretations.

**Author's response**
We have completely reworked and rewrote the discussion section. We dropped the separation of the three sites and grouped the interpretations/discussion related to our sites.

**Author's change in manuscript**
See the new discussion section, which is now structured as such: 5.1. Initial landform development. 5.2. Differences between debris-covered and rock glacier areas. 5.3. Current evolution and its significance. 5.3.1. Landscape evolution. 5.3.2. Dynamical evolution. 5.3.3. Final diagnostics and future evolution of the landforms.
* * *
**Comment from referee, # 6**
For instance, you mention compression and bulging in chapter 4.2.4. How do you reach these conclusions? This is unclear.

**Author's response**
This is one challenge associated with interpreting measured elevation changes and horizontal displacements in terms of flow mechanism. This challenge has been dealt with before, for example by Lambiel and Delaloye (2004), and we have followed their guidelines for interpreting our data, as outlined in the Methods section of the initial manuscript version. See p. 9, l. 11-25: "As outlined by Lambiel and Delaloye (2004), elevation changes at the surface of rock glaciers may be explained by several and possibly concomitant factors: downslope movement of the landform and advection of local topographic features, extensive or compressive flow, and melting or aggradation of internal ice. Therefore, it is difficult to unambiguously interpret elevation changes. Studying the Muragl rock glacier (Swiss Alps), Kääb and Vollmer (2000) highlighted how mass advection caused subtle elevation changes (between $-0.20$ and $+0.20$ m yr$-1$), while surface lowering of up to $-0.5$ m yr$-1$ were considered as indicative of massive losses of ice. Accordingly, taking into account the range of values measured and the uncertainty (or detection threshold) on the measurements (see Table 2), we used an absolute value of 0.50 m yr$-1$ to generally discriminate between 'moderate' and 'large' vertical changes. The former were considered to relate primarily to the downslope expansion of the landform (including long profile adaptation and advection of topographic features) and, thus, to extensive flow; in the case of the latter, additional ice melting or material bulging by compression were considered necessary in the interpretation." Furthermore, in the new version of our work we used the streamlines (interpreted from vector displacement maps) in order to better interpret the altitudinal changes.

**Author's change in manuscript**
See the interpretations related to Results.
* * *
**Comment from referee, # 7**

Another example: You use the permafrost probabilities given by your modelling for the interpretation of spatial differences on case one. This seems simplistic: a model is not the truth, especially with permafrost spatial models. The reality is often much more complex at the site scale.

**Author's response**
Of course, model results do not replace observations. However, the permafrost probalistic model relies on spatial differences in potential incoming radiation and elevation, which are the primary controls on the surface energy balance and hence of permafrost occurrence. We are confident that the model expresses, at the very least, the first order spatial variations in microclimatic conditions and so is useful to interpret spatial variations in the morphological, topographical and displacement data obtained in this study. Nevertheless, we have lowered its importance in the interpretation section of the manuscript.

**Author's change in manuscript**
See Results and Discussion, we now speak generally of 'favourable topoclimatic conditions' (which include the permafrost models cited in Study sites).
* * *
**Comment from referee, # 8**
Honestly I do not see any clear difference between Navarro and Las Tetas regarding glacier – rock glacier interactions. How can you conclude that there was only a colliding between the glacier and the rock glacier in Las Tetas, and no superimposition by the glacier, as it is the case in Navarro? What about the origin of both rock glaciers? Are they glacier-derived rock glaciers, or talus rock glaciers overridden by a glacier? You should discuss this.

**Author's response**
Indeed, their initial development may have been analogous and we acknowledge that the differences between both were not correctly exposed in the first manuscript version. Now, we insist on the fact that Navarro is a much larger and more complex landform than Las Tetas and that their current evolution is radically different. In particular, the Navarro's eastern unit is quite similar to Presenteseracae. For the western unit, see the change cited below.

**Author's change in manuscript**
See the whole Discussion section, and in particular subsection 5.4.: "The Navarro's western unit and Las Tetas are more commonly known cases of assemblages that have formed and evolved in reaction to the superimposition/embedding of glaciers onto or in the back of rock glaciers and their subsequent dynamical interactions (see Introduction: type [i]). In both cases, the progression of the rock glacier at the expense of the debris-covered glacier is rather limited (Navarro's western unit) or null (Las Tetas). At Navarro's western unit, the debris-covered glacier has displaced more slowly than the rock glacier. At Las Tetas, there is no major contrast in displacement speed between the debris-covered and the rock glacier. In both cases, the better depiction of flow patterns and their more extensive nature evoke a 'locomotive' role for the rock glacier. It is nevertheless difficult to assert whether the debris-covered glaciers are 'pushing away' the rock glaciers or if the latter are 'pulling' the former; both processes probably occur. The dynamical links between both units certainly constitutes a complex issue deserving more attention. The flow dynamics highlighted at Navarro supports the idea of a close relationship between the debris-covered and rock glacier, with

many streamlines connecting both areas. At Las Tetas it is very conspicuous how during the studied period the debris-covered and the rock glacier changed from being dynamically coupled to essentially de-coupled after 1978. Furthermore, as these whole landforms continue to advance, the rock glaciers could plausibly become entirely isolated from their main debris source in the upper cirques while the increasingly warming conditions could cause the debris-covered glacier to become stagnant or disappear. Also, as the rock glaciers penetrate in areas with less favourable topoclimatic conditions, their future sustainment can be questioned."
* * *
**Comment from referee, # 9**

However, you must be very cautious with the interpretation on rock glacier origin, since you have no geophysical data to show the internal structure.

**Author's response**

We agree to the somewhat tentative nature of these interpretation with the data at hands. Nevertheless, we do not think that geophysical data always give the necessary "clues" to the landform origin. First, they also need interpretation and the latter can be delicate and disputable. Second, in the best case, geophysical data allow for ice content estimates/reconstructions. In an absolute way, the ice content of a rock glacier does not tell anything about its origin. For example, Monnier and Kinnard (2016) have demonstrated that very large (>2 km long) rock glaciers with high ice contents (>70-80%) and located in glacially shaped areas can, from a physical point of view, perfectly develop by pure periglacial processes.

**Author's change in manuscript**

No change made.
* * *
**Comment from referee, # 10**

The references are rather few.
You must much more discuss your findings regarding previous studies.

**Author's response**

We added references (in total 14) when needed, especially in the interpretations developed in the Discussion section. Nevertheless, note that we also depict and discuss phenomena that were never previously reported, such as the replacement of debris-covered glacier morphology areas by rock glacier morphology areas.
* * *
**Comment from referee, # 11**

p.1, l.18. 'monitored' is not suitable here. Prefer 'investigated', 'assessed', …

**Author's response**

OK

**Comment from referee, # 12**
p.2, l.9-18. Please add references.

**Author's response**
Done.

**Author's change in manuscript**
See the introduction: "The most striking geomorphological expression of glacier–rock glacier interactions are large glacier–rock glacier transitional landforms which are assemblages of debris-covered glaciers in their upper part and rock glaciers in their lower part (e.g., Kääb et al., 1997; Krainer and Mostler, 2000; Ribolini, 2007; Monnier et al., 2014; Janke et al., 2015). Here, it is important to recall and highlight the differences between both types of landforms (Nakawo et al., 2000; Kääb and Weber, 2004; Cogley et al., 2006; Haeberli et al., 2006; Degenhardt, 2009; Benn and Evans, 2010; Berthling, 2011)." And further.

**Comment from referee, # 13**
p.2, l.11. The debris cover can be continuous.

**Author's response**
Yes, it is right. We subtly modify the sentence.

**Author's change in manuscript**
p. 2, l. 27-28: "Debris-covered glaciers are glaciers covered with a thin (no more than several decimetres thick) and *generally discontinuous* debris layer."

**Comment from referee, # 14**
p.2, l.14. "RG morphology is coherent, stable, …" What is a coherent and stable morphology? Please be more accurate.

**Author's response**
The reviewer is perfectly right in asking more precisions about this terminology. The use of the term 'coherent' in our works (see also Monnier and Kinnard, 2015) relates mainly to the consideration and citation of the dissertation by Berthling (2011: "Beyond confusion: rock glaciers as cryo-conditioned landforms"). In his article Berthling employs the term 'coherent' in order to describe the surface morphology of rock glaciers and especially differentiate it from that of debris-covered glaciers, e.g. (p. 3): "For a general glacier to rock glacier transformation model to work, as an alternative to the permafrost model, there are some criteria that must be met. First, the presence of permafrost must be ruled out. Second, measurements or modelling must show that the melting of the ice core would not proceed at a rate higher than what would allow for a coherent surface morphology to be maintained and developed." Here, 'coherent' obviously refers to 'stable and organized' (lobes and ridge-and-furrow patterns). The term (in Berthling's lines as well as in ours) is also probably an indirect reference to the frequent use of the term 'cohesive flow' in the rock glacier literature. Barsch (1992) directly refer to cohesive flows in his definition of rock glaciers (p. 176): "Active rock glaciers are the visible expression of steady-state creep of supersaturated mountain bodies

of unconsolidated materials. They display the whole spectrum of forms created by cohesive flows."

'Cohesive flow' is an expression used in sediment rheology in order to characterize hyper-concentrated debris flows or turbid currents and alludes to the 'cohesion strength' of the flow (e.g., Hsü, 2004). The notions of cohesion and cohesive force also appear in the lava rheology literature (e.g., Manga and Ventura, 2005). Here, we will retain the essential idea that the creep of rock glaciers is a flow of sediments made cohesive by the presence of ice and permafrost which produces peculiar features (mainly the ridge-and-furrow pattern) on the surface; in the literature these features have been many times refers to as 'viscous' because they are clearly comparable or in some contexts (e.g., volcanic high mountain) confusable with the features appearing at the surface of hyper-concentrated debris flows or lava flows.

In the new version of our manuscript we have decided to drop the use of the term 'coherent', which seems ambiguous, and to use the following terminology in order to describe and identify rock glacier morphology: 'stable', 'organized', 'organized features', or 'cohesive flow-evocative features.

**Author's change in manuscript**
See Introduction (p. 1, l. 1-10): "On their surface, rock glaciers exhibit "the whole spectrum of forms created by cohesive flows" (Barsch, 1992, p. 176) of "lava-stream-like (…) viscous material" (Haeberli, 1985, p. 92). These features vary upon the case and study area; according to our field surveys in the Andes, they can be grouped in three main types: small-scale (<1 m high) ripples or undulations resulting from deformations in the active debris layer moving together with the underlying perennially-frozen core; medium-scale (1−5 m high) ridge-and-furrow assemblages resulting from the compression in the whole ice−debris mixture; and large scale (5−20 m thick and >100 m long) superimposed flow lobes upon which the first two types may naturally appear. Hereafter, we will simply refer to these features as 'cohesive flow-evocative features'."
* * *
**Comment from referee, # 15**

Fig. 1: Explain in the text the significance of the drainage network. In addition, it appears violet instead of blue on the map.

**Author's response**
The interest of the drainage network on this figure is to illustrate the variations of climatic−hydrologic conditions along Chile. We have precised it in the caption. Also the color has been changed.

**Author's change in manuscript**
"Figure 1. Location of the study sites. Drainage network, *which reflects the variations of climatic−hydrologic conditions along the Chilean territory,* is shown in blue."
* * *
**Comment from referee, # 16**

p.4, l.25 and Fig.3. It is unclear how the limit between the rock glacier parts and debris-covered parts were drawn. On Fig. 3 you should draw the limits of the Western and Eastern units.

**Author's response**
Limit between rock glacier and debris-covered glacier parts: see our answer to Comment #4. The eastern and western units are clearly delimited by a series of central morainic crests, as precised in Study sites.
* * *
**Comment from referee, # 16**
Fig. 4: Indicate the location of Las Tetas on Fig. 3. Indicate by arrows or signs the location of the central depression and of the thermokarst on the photos.

**Author's response**
Las Tetas is 300 km north from Navarro therefore there would be a scale problem with this requirement.

**Author's change in manuscript**
The central depression and the thermokarst areas have been indicated on Figure 4.
* * *
**Comment from referee, # 16**
p.5, l.10-13. These lines are a bit fuzzy. The model used is only indicative and, in my opinion, cannot be used to explain in a so simple way the morphological differences encountered on the different parts of the landform. In addition, I observe that thermokarst is also present were permafrost probability is higher than 0.95.

**Author's response**
Absolutely, the model is only indicative, we did not aim at using it in order to simplistically explain the landform organization. We have rephrased in order to avoid any ambiguity.

**Author's change in manuscript**
"Monnier and Kinnard (2015) provided an empirical model of permafrost probability based on logistical regression for the upper Aconcagua River catchment. According to this model, Navarro may be in a permafrost state. The permafrost probability is close to 1 in the upper parts, nevertheless there is a marked decreasing gradient in permafrost probability from 0.9 to 0.7 between the central part and the terminus of the western unit."
* * *
**Comment from referee, # 17**
p.5, l.28-32: In my opinion this is not really a rock glacier morphology, because the landform is very thin, especially the northern lobe and because the surface ridges are not classical compression ridges of block-ice mixture, but rather ridges due to subsurface ice flow.

**Author's response**

We disagree with the use of the rock glacier thickness in order to identify rock glacier morphology. The aim is to examine what kind of morphology is present *on the surface*: is it organized, evocative of a cohesive flow? Features that fulfil this requirement can be of various types on rock glacier surfaces. See Introduction and our answer to Comment #14. We solely aim at using the presence of cohesive flow-evocative features in order to highlight rock glacier areas. At last, the discussion of the status of appearing rock glacier for Presenteseracae has been previously thoroughly examined and discussed (Monnier and Kinnard, 2015) and will not be repeated here.
* * *
**Comment from referee, # 18**
p.5, l.29-30: "Small morainic": I do not understand what you are talking on.

**Author's response**
"Small morainic crests" was not the appropriate term. We replaced it by the more accurate 'push moraine' referring to the terminology by Benn and Evans (2000). These features mark the receding of the Presenteseracae glacier during the last decades-century.
* * *
**Comment from referee, # 19**
p. 6 l. 1: remove 'ice'. l. 22: 100 m high seems to be exaggerated. l. 27 and subsequent: "aerial photos" is more appropriated than 'air photo'.

**Author's response**
The suggested corrections were done. Regarding the height of the Las Tetas front, it corresponds to the height of the frontal talus, which obviously may bury sediments or outcrops downward: it is 100 m. Nevertheless, we have modified the text in order to avoid any misunderstanding.

**Author's change in manuscript**
p. 7, l. 29-30: "The front of Las Tetas is prominent; its talus slope, which may bury sediments or outcrops downward, is almost 100 m high (Fig. 4)."
* * *
**Comment from referee, # 20**
p.8, l.1. Prefer "altitudinal changes" to "vertical displacement". Valid for the rest of the text.

**Author's response**
We have preferred 'elevation changes', as suggested by the other reviewer.
* * *
**Comment from referee, # 21**
p.8, l.14. There is an issue with this interpretation of vertical changes. If a vertical change occurs locally, this means that an additional process to downslope movement occurs. Indeed, if only downslope movement occurs (for instance without ice melt, ice aggradation or compression), then the altitude at the same location will not change.

**Author's response**

The latter assertion is wrong. Vertical elevation changes result from the integrated mass fluxes to the column, which include flow divergence or convergence and ice melt or debris/ice accumulation. As outlined in this method subsection, and following, e.g., Lambiel and Delaloye (2004), downslope movement can produce a negative or positive change of altitude: (i) the longitudinal geometry changes, with a terminus at a lower elevation and thus, for a given geometry, and especially in the generally concave topographies where rock glaciers occur, a longitudinal profile that will stand at lower elevations; (ii) the downslope movement of the rock glacier can translate downward local positive (e.g., lobe front) or negative (e.g., depression) topographic features, which will result in an altitude change.
* * *
**Comment from referee, # 22**

p.8, l.19. You must explain how this boundary was defined. Which criteria were used to distinguish rock glacier morphologies from debris-covered ones? This is important since a large part of the study relies on this distinction.

**Author's response**

See Comment #4.
* * *
**Comment from referee, # 23**

p.9, l.3. As far as I know, it is not necessary that the two images must be taken from the same position. With remote sensed images it is even impossible.

**Author's response**

Absolutely, we have removed that unnecessary part of the sentence.
* * *
**Comment from referee, # 24**

p.9, l.15-20 and Tab. 3. It is unclear for me how the search window was defined: what is the relationship between columns 4 and 5 in Tab. 3? In addition, the maximum expected displacement can be well above the front displacement.

**Author's response**

There was indeed a lack of precision regarding this methodological aspect. The search window (col. 5) was defined sufficiently large compared with the maximum front displacement expected (col. 4): we took the latter value in pixels, multiplied it by 2 and rounded at the upper common search window size. The caption of the table has been completed. This was done in order to foresee what the reviewer says, i.e., that the maximum expected displacement can be well above the front displacement.

**Author's change in manuscript**

p. 10, l. 28-p. 11, l. 2: "Then for each time interval the maximum front displacement was measured and used for defining the search window size accordingly: we converted the maximum front displacement in pixels, multiplied the value by 2, and chose an upper

rounded value; hence, being sufficiently large compared to the maximum front displacement the search window could track potential larger displacements on the surface (Table 3). The precision of the measurement of the maximum front displacement was estimated to be ±5 m taking into account the errors related both to orthorectification and the mapping error on the images."
* * *
**Comment from referee, # 25**

p.10, chap. 4.1.1. Without viewing the aerial images, it is impossible to see how you reached your conclusions about geomorphological changes. You give a change value of 75%, but how can you make a spatial quantification of the changes, knowing that differences between rock glacier and debris-covered glacier morphologies are somewhat not well defined and probably quite subjective.

**Author's response**

We have now included a figure with the aerial photos and satellite images at each site for the reader to interpret. The spatial quantification is here an approximation of the areas that appear from each side of the rock glacier/debris-covered glacier morphology boundary. Since the distinction is based on precise criteria as exposed previously, we deem this first-order quantification to be satisfactory.

**Author's change in manuscript**

Fig. 6.
See also p. 12, l. 6-9: "At Navarro, rock glacier morphology areas have progressed spatially between 1955 and 2014, especially in the eastern unit where they now approximately represent three quarters of the total area; the boundary between debris-covered and rock glacier morphology areas has progressed upward considerably (~400 m) between 1955 and 2014 (Fig. 6)."
* * *
**Comment from referee, # 26**

p.10, l.20. What is this 'current morphology'?

**Author's response**

The sentence was "this reduction has to be related to the progression of the coherent morphology from the margins of the feature inward". As precised before, we have dropped the use of the term 'coherent morphology'.
* * *
**Comment from referee, # 27**

p.10, chap. 4.1.3 and Fig. 6&7. As already mentioned it is very difficult to read the velocities. In addition, the directions appear to be very noisy, and a systematic error seems to occur, as indicated by the general south-west direction.

**Author's response**

See our General comment dealing with the displacement velocity problems and the new figure and colour scales.

**Comment from referee, # 27**

p.10, l.27. What are exactly the displacement rates for rock glaciers? The values given by Barsch and Haeberli may be true for a period running up to the early 2000's, but now several studies have shown that the velocities have strongly increased in the recent years, and that velocities above 1 m/y is very often the norm.

**Author's response**

Indeed, several studies have highlighted increases in rock glacier speeds during the last 20 years. We included a recent study in the references. Nevertheless, despite that acceleration, average velocities on an order of magnitude of several dm/yr remain 'typical', i.e., neither very slow nor very fast.

**Author's change in manuscript**

See p. 14, l. 10-12: "Displacements are one order of magnitude higher than those on the surface of Navarro and appear as relatively high for rock glaciers (see reference values in Barsch, 1996; Haeberli et al., 2006; Scapozza et al., 2014)."

**Comment from referee, # 28**

p.11, l.17. I do not see the heaving in the fronts and the margins.

**Author's response**

This is right: the conclusion was inappropriate and we removed it.

**Comment from referee, # 29**

p.11, l.18. See my comments concerning the Method (interpretation of vertical movements).

**Author's response**

See our General comment.

**Comment from referee, # 30**

p.11, l.27-30. This evolution is very difficult to see on the images. Could you make a zoom on the frontal part? In addition, try to better show the rock glacier morphology (in Fig. 4_3 we do not see the front).

**Author's response**

Since details relative to this site can be found in Monnier and Kinnard (2015), we have preferred not to develop furthermore in order not to increase the number of figures which are already numerous. Nevertheless, see Fig. 10a-11b where we have tried to depict more precisely the surface feature details.

**Comment from referee, # 31**

p.12. chap. 4.2.3. Obviously there are no velocity differences between the debris-covered glacier part and the rock glacier part. Is this realistic? Again, are you sure that there is no problem with the method? l.17-18. I do not see the two flow lobes in the upper part.

**Author's response**

See our General comment, the new figures, and in the discussion subsection 5.2. "Differences between debris-covered and rock glacier areas." Also 5.3.2. Dynamical evolution. Generally, our study shows that the dynamical difference between debris-covered and rock glacier areas lies much more in flow pattern organisation than in velocity values. Nevertheless, at Navarro, we note a clear increase in displacement speeds from the upper debris-covered to the lower rock glacier part (see subsection 4.1).

**Author's change in manuscript**

See the whole subsection 5.2. (p. 17, l. 31-p.18, l.14): "Our study basically relied on the landscape differentiation between debris-covered and rock glacier areas. The criteria enounced and discussed in the Introduction section have been used to distinguish and partition the surface morphology of the landforms studied. Our subsequent results show that, at Navarro and Las Tetas, debris-covered and rock glacier areas are characterized by contrasting horizontal displacement and altitudinal change patterns. Flow patterns in rock glacier areas are conspicuous and express the cohesive downslope expansion of the landform in the direction of its main axis; in terminal and marginal areas, significant to pronounced bulging occurs by compression. Flow patterns in debris-covered glacier areas are generally less conspicuous and express the interference of instability processes, in particular ice melt-governed downwasting, with downslope expansion. In particular, altitudinal changes in debris-covered glacier areas are stronger in amplitude and more spatially contrasted. As mentioned before, the lower density and definition of displacement vector patterns and corresponding streamlines in the debris-covered areas may explain by inherently less cohesive mass flow. These whole flow dynamics appear perfectly coherent with the definition of and distinction made between debris-covered and rock glaciers in the Introduction section."
* * *
**Comment from referee, # 31**

p.12. chap. 4.2.4. Your conclusions regarding compression, bulging, expansion and ice melt are too direct and lack any demonstrations. How can you conclude to these processes? This must be better explained and discussed. For instance, at the front, the vertical gains may also be attributed to the rock glacier advance and not by compression: in 2000 the rock glacier front was (?, you can verify) located upside the current position, so the blue colors here would just indicate the downslope progression of the front.

**Author's response**

See our answer to Comment #6. We have tried to be more cautious in the interpretation of altitudinal changes in this new version.

**Author's change in manuscript**

p. 14, 23-29: "In detail, one will also notice the occurrence of streamlines converging towards one another or towards depressed topographic features: either at the location of surface heaving (for both periods, see around 3900 m asl, in the upper part of the landform), which may highlight the (temporary?) formation and distal bulging by compression of individual flow lobes; or at the location of pronounced surface lowering (lower southern part between 2000 and 2014), which highlights the interference of ice losses-related downwasting with extensive flow."
* * *
**Comment from referee, # 32**

p.13, chap. 4.3.3. The vectors are unreadable

**Author's response**

They were indeed unreadable. We have considerably reworked the figures and artwork and hope that the resulting new figures offer better reading conditions.
* * *
**Comment from referee, # 33**

p.15, l.29-32. I don't understand this sentence.

**Author's response**

This sentence is not present anymore in the manuscript.
* * *
**References (not cited in the manuscript)**

Hüs, K.J.: Physics of sedimentology. Springer, Berlin, 2004.

Manga, M., Ventura, G. (Eds.): Kinematics and dynamics of lava flows. The Geological Society of America, Boulder, 2005.

Monnier, S., Kinnard, C.: Interrogating the time and processes of development of the Las Liebres rock glacier, central Chilean Andes, using a numerical flow model. Earth Surface Processes and Landforms, in press, 2016. http://onlinelibrary.wiley.com/doi/10.1002/esp.3956/abstract

Nuth, C., Kääb, A.: Co-registration and bias corrections of satellite elevation data sets for quantifying glacier thickness changes. The Cryosphere, 5, 271−290. http://www.the-cryosphere.net/5/271/2011/tc-5-271-2011.pdf

---

## Author Response (AR2)

**Associate Editor Decision**

Thank you for a much-improved revised manuscript. I attach a version in which I have annotated some punctuation suggestions which I would like you to include if you decide for a resubmission. As you will see from the commissioned referee reports there remain questions of your data reliability and methodology reproducibility which you will want to address before we can consider your manuscript for final publication. I hope that you will be able to address these two concerns and forward us a revised version for final appraisal for inclusion in Earth Surface Dynamics.

*Answer: We have completely reprocessed the data using another program and used new pre-processing of images (contrast enhancement and snow masking) along with more stringent filtering criteria in order to address the two abovementioned concerns (see below and new Methodology Section). The paper has been modified accordingly, with important changes to the figures and tables, and corresponding changes to the text where needed to accommodate the updated results; the core of the interpretations and main insights drawn in the Discussion section remain quite unchanged. Most modifications to the interpretations concern the updated results on the surface displacements obtained from the new feature tracking analysis. We have furthermore taken into account the punctuation and language suggestions provided by the Associate Editor for this new version. We have included as uploaded files a revised version of the manuscript (V3) were the main changes are highlighted using the MS Word change track function, and a clean copy of this latter file.*

**Referee #1 (Prof. Dr. Andreas Kääb)**

General comment

The paper looks much improved. However, at this stage I looked in detail only at the displacement vectors as they were a key critic from my side. I am still quite sure that there is a major problem with the displacement measurements.

I copied the orthophotos of the different years and different rock glaciers from the pdf and overlaid and flickered them. In the lower parts of the rock glaciers there clearly is coherent movement that could be measured. In the upper parts, though, I doubt that most of the displacement vectors shown in the paper are true, especially not for the longer time periods investigated. Many of the vectors are on places which either have snow in one of the images, or where heavy thermokarst processes seem to alter the surface. Under such circumstances, I cannot imagine how image correlation should be able to provide displacements, even if only low-res versions of the air photos are available to me for my assessment.

*Answer: We finally totally acknowledge that in the previous versions of our MS, issues occurred relating with the processing of images with CIAS. The whole analysis was re done completely, using more careful pre-processing of the images and more stringent filtering criteria. This included masking out snow-covered surfaces, image contrast enhancement, adapting the template window according to the image quality and the spatial scale of surface morphology expected to be preserved over time, and application of a 3-criteria filtering procedure to the output vectors matched by the NCC algorithm. As detailed in the updated Methodology section, we have decided to re-process entirely the images using the MATLAB toolbox ImGRAFT, with which the second author had extensive experience. The updated analysis resulted in much less matches as well as the removal of blunders and/or too chaotic patterns, which were correctly detected by A. Kääb (especially those related to snow cover...). The new results are nonetheless interesting and instructive, with spatially coherent displacement vector patterns, indicative of slow creep, being most consistently found over rock glacier areas, whereas debris-covered areas were rather characterized by either a lack of movement detection and/or chaotic, poorly organized flow patterns evocative of thermokarst degradation processes. The rest of the paper has been modified according to these new results.*

**Referee #2**

**General comment on the revised version**

The manuscript has been substantially improved since the first version. All the questions have been satisfactorily answered. The manuscript is now much better structured, the discussion much more consistent and the significance of the study clearer.

I have just a problem with the streamlines. It is not clear how they were drawn, and so their mapping appears to be highly subjective. There are probably hundreds of ways of drawing the streamlines. As they obviously come from the displacement vectors, the same analyse could somehow be done from the displacement vectors. Thus, how the streamlines were drawn must be much better explained, because the method must be reproducible by others. Otherwise you may also consider removing the streamlines.

*Answer: Following the modifications related to the methodological changes (see responses to Referee #1), we have decided to remove the streamline delineation from this study. The updated velocity fields did not allow depicting the streamlines for each landform and periods, as in some cases only limited movement was detected on the surface*

**Other comments:**

P1, l28. Prefer « evolved » to « transformed »
*Answer: OK.*

P3, l23. Remove « That being said, »
*Answer: OK.*

Fig. 3. The limit between the Western and Eastern units are still missing.
*Answer: We have added a dotted line on the figure in order to separate both units.*

P6, l26. Replace « well-defined steep frontal talus slopes » by « a well-defined steep front », because a talus slope is an accumulation of debris at the foot of a rock wall. This is not the case here. Same remark for p7, l29.
*Answer: OK.*

Legend Fig. 7. The front of which landform? Debris-covered glacier I guess, so please give precision. What is the black line?
*Answer: The front of the studied landform. Considering that the latter has evolved over the studied period from a single debris-covered glacier to a currently forming rock glacier, we prefer not give to any precision.*

P12, l7. « …where they now approximately represent three quarters… ». How can you give this estimation since the western and eastern units are not mapped?
*Answer: We have dropped such approximate precision in the new version of the manuscript.*

P12, l13-15. Again, we want to see a mapping of the thermokarst areas.
*Answer: Thermokarst areas are mapped on Fig. 9−14 when present and detectable.*

P14, l3-5. Why thermokarst would be controlled by the glacier size? Maybe there is no thermokarst because of the steep slope?

**Answer:** *Glacier size increase the area and volume of materials (debris and even more ice), morphological pattern diversity, and, subsequently the probability of encountering thermokarst. Nevertheless, indeed, the slope may play here a role in the absence of thermokarst. We modified the sentence accordingly.*

[revised manuscript text omitted]